# CGLB: Benchmark Tasks for Continual Graph Learning

**Xikun Zhang**
The University of Sydney
xzha0505@uni.sydney.edu.au

**Dongjin Song**
University of Connecticut
dongjin.song@uconn.edu

**Dacheng Tao**
The University of Sydney
dacheng.tao@gmail.com

## Abstract

Continual learning on graph data, which aims to accommodate new tasks over newly emerged graph data while maintaining the model performance over existing tasks, is attracting increasing attention from the community. Unlike continual learning on Euclidean data (*e.g.*, images, texts, etc.) that has established benchmarks and unified experimental settings, benchmark tasks are rare for Continual Graph Learning (CGL). Moreover, due to the variety of graph data and its complex topological structures, existing works adopt different protocols to configure datasets and experimental settings. This creates a great obstacle to compare different techniques and thus hinders the development of CGL. To this end, we systematically study the task configurations in different application scenarios and develop a comprehensive Continual Graph Learning Benchmark (CGLB) curated from different public datasets. Specifically, CGLB contains both node-level and graph-level continual graph learning tasks under task-incremental (currently widely adopted) and class-incremental (more practical, challenging, yet underexplored) settings, as well as a toolkit for training, evaluating, and visualizing different CGL methods. Within CGLB, we also systematically explain the difference among these task configurations by comparing them to classical continual learning settings. Finally, we comprehensively compare state-of-the-art baselines on CGLB to investigate their effectiveness. Given CGLB and the developed toolkit, the barrier to exploring CGL has been greatly lowered and researchers can focus more on the model development without worrying about tedious work on pre-processing of datasets or encountering unseen pitfalls. The benchmark and the toolkit are available through https://github.com/QueuQ/CGLB.

## 1 Introduction

In real-world applications, many graph data are generated continuously. For instance, in a citation network, new types of papers and their associated citations may emerge and an ideal document classifier needs to continuously adapt its parameters to distinguish the documents of newly emerged research fields [31, 63, 64]; for drug design, new categories or properties of molecules may be encountered and a molecule property prediction model will need to keep updating its parameters to learn new molecule categories or properties [31]. Therefore, given either a continuously expanding graph (with new types of nodes and associated edges) or new categories of graphs continuously emerging, for practical concerns, it is important to develop Continual Graph Learning (CGL) that can accommodate new tasks over newly emerged graph data while maintaining the model performance

36th Conference on Neural Information Processing Systems (NeurIPS 2022) Track on Datasets and Benchmarks.

over existing tasks. However, unlike continual learning on Euclidean data (*e.g.*, images, texts, etc.) that has established benchmarks and unified experimental settings [30, 36, 45, 61, 20, 37, 28, 15], CGL is still immature and with very rare public benchmark tasks being available. Moreover, due to the variety and complexity of graph data, different datasets have been adopted by existing works with different experimental settings. This causes great difficulties in model comparisons and hinders the development of this field. Therefore, it is urgent to create benchmark tasks curated for CGL with standard settings for different graph datasets to minimize the effort spent on tedious data curation as well as the configuration of continual learning tasks, and to avoid encountering potential pitfalls.

In this paper, we first provide a systematic taxonomy of different CGL scenarios that include (1) node-level CGL (N-CGL), which deals with node-level prediction on a single growing graph with new types of nodes continuously emerging; and (2) graph-level CGL (G-CGL), which deals with graph-level prediction with new categories of graphs continuously appear. Moreover, we clarify the difference between the currently widely adopted task-incremental learning (task-IL) setting and the underexplored yet more challenging class-incremental learning (class-IL) setting, and point out the proper application scenarios of these two settings. Based on the established taxonomy, we construct the Continual Graph Learning Benchmark (CGLB) with benchmark tasks for each setting. Along with CGLB, we provide a comprehensive toolkit for training, evaluation, and visualization of different CGL models, which could greatly alleviate the burden/risks to investigate CGL problems. Finally, with CGLB and the toolkit, we conduct comprehensive studies on the existing state-of-the-art methods, which reveal the effectiveness of these methods in different scenarios and point out challenges of the existing models under certain settings. With all these efforts, we not only lower the barrier to explore the CGL problem but also provide a systematic overview of the current situation of CGL and point out several challenging directions for future studies.

## 2 Related Works

### 2.1 Continual learning

In the past, continual learning has been explored in many areas including computer vision [19, 51], reinforcement learning [38], *etc.* Essentially, it targets at resolving the catastrophic forgetting problem, *i.e.* the drastic decrease of a model's performance on previous tasks after being trained on new tasks. Existing works on continual learning can be categorized into three types, *i.e.*, regularization-based methods, memory-replay based methods, and parametric isolation based methods. Regularization-based methods seek to maintain the model performance on previous tasks by penalizing the changes in the model parameters via regularization terms [23, 29, 25, 12, 40]. For example, Kirkpartick *et al.* [25] introduced elastic weight consolidation (EWC) to prevent the model parameters from shifting too much via a quadratic penalty. Several recent works [12, 40] learn the gradients for new tasks within a subspace that is orthogonal to the gradient directions that increase the loss on previous tasks. Memory-replay based methods select a set of representative data from previous tasks, which are used to retrain the model with the new task data to prevent forgetting [32, 42, 4, 6, 9, 26]. For instance, Gradient Episodic Memory (GEM) [32] clips the gradient of the current task to prevent the loss of the data in the episodic memory from increasing. Shin *et al.* [42] avoided directly storing data and instead used a generative model to obtain pseudo-data of previous tasks . Better designs of memory replay are also explored recently [6, 9] since it is simple yet effective. Parametric isolation based methods avoid drastic changes to the parameters that are important to previous tasks by introducing new parameters for new tasks [39, 59, 58, 50, 52]. Progressive network [39] avoids modification to the networks for previous tasks by allocating a new sub-network for each new task. Various new approaches in this branch are also developed recently [52, 58, 50]. With the prosperity of continual learning, there are also a number of benchmarks and studies on the continual learning settings [30, 36, 45, 61, 20, 37, 28, 15].

### 2.2 Continual learning on graphs

Recently, continual graph learning (CGL) is attracting significant amount of attention from the community due to its practical importance [47, 55, 10, 27, 1, 7, 46]. These works can also be categorized as regularization methods (*e.g.*, Topology-aware Weight Preserving (TWP) [31]) that preserves crucial topology for previous tasks, parametric isolation approaches (*e.g.*, Hierarchical Prototype Networks (HPNs) [63]) that select different parameters for different tasks, and memory replay

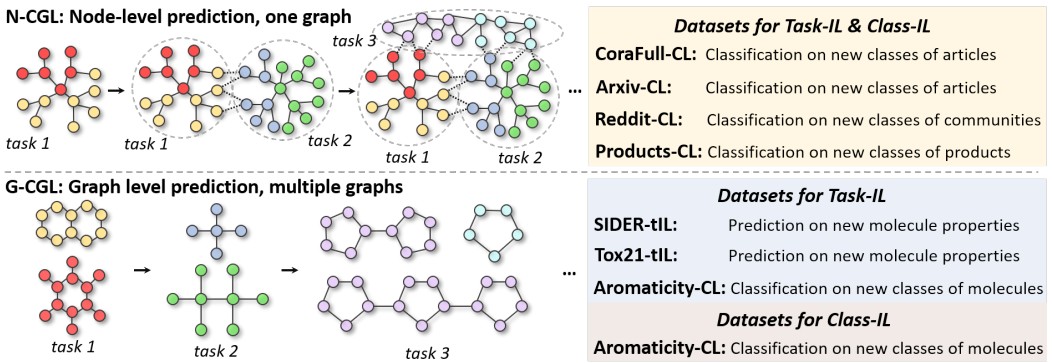

Figure 1: Overview of 4 different benchmark tasks with N-CGL or G-CGL setting under task-IL or class-IL scenario (applied to different datasets).

methods (*e.g.*, Experience Replay Graph Neural Network (ER-GNN) [64]) that stores representative nodes. However, to the best of our knowledge, there is only one existing work [8] on benchmarking CGL and no study has been conducted over the different settings of CGL tasks. Above all, it is necessary to develop comprehensive benchmarks of both node-level and graph-level tasks for CGL with different learning scenarios across multiple datasets. Due to the absence of public benchmarks for CGL, existing works adopt different protocols to configure datasets and experimental settings. Different configurations have their own rationales tailored for specific scenarios, but without a unified framework, it is difficult to fairly compare different techniques and thus hinders the development of CGL. In this work, we systematically clarify the most important settings including both node-level and graph-level prediction tasks under both task-IL and class-IL scenarios, and construct the Continual Graph Learning Benchmark (CGLB) with benchmark tasks for each setting.

Finally, the essential difference between CGL, dynamic graph learning [14, 48, 18, 60, 35, 66, 33, 13, 5], and few-shot graph learning [65, 16, 57, 43] is worth noting. Dynamic graph learning focuses on capturing the temporal graph dynamics and keeping the graph representations up to date (instead of focusing on the forgetting problem), with access to all previous information. In contrast, CGL targets the forgetting problem, therefore the data from previous tasks are inaccessible. Note that an exception is the CGL with inter-task edges (introduced in Section 3.1), which allows the information from previous tasks to be aggregated via the inter-task edges in the neighborhood aggregation operation of GNNs. But the labels of the data from previous tasks are still inaccessible. Few-shot graph learning aims at fast model adaptation to new tasks. In the training, few-shot learning models have access to all tasks simultaneously, which is not available in CGL (CGL with inter-task edges is slightly different, as introduced in Section 3.1). During the evaluation, a few-shot learning model is tested on new tasks, on which the model is first fine-tuned, while the CGL models are tested on existing tasks without any fine-tuning.

## 3 Continual Graph Learning Benchmark

In Continual Graph Learning (CGL), a model is expected to learn on a sequence of tasks during the training phase. After learning all tasks, the model is evaluated on each learnt task individually to see how effective the model is on preventing forgetting over existing tasks. Unlike classical continual learning with independent data instances (*e.g.*, images and texts), CGL can be categorized into node-level CGL (N-CGL) and graph-level CGL (G-CGL) settings as shown in Figure 1. Tasks in N-CGL are node classification on a single expanding graph with different tasks containing different node classes. In this case, different nodes are connected by edges and thus not independent. Moreover, when new types of nodes emerge, edges connecting different tasks also appear simultaneously, which is a unique problem in CGL compared to classical continual learning. The G-CGL, on the other hand, resembles the classical continual learning more, since the examples are individual graphs and without interconnected edges. In this section, we will focus on introducing the main ideas of different CGL settings. More details are included in Section 1 of Appendix.

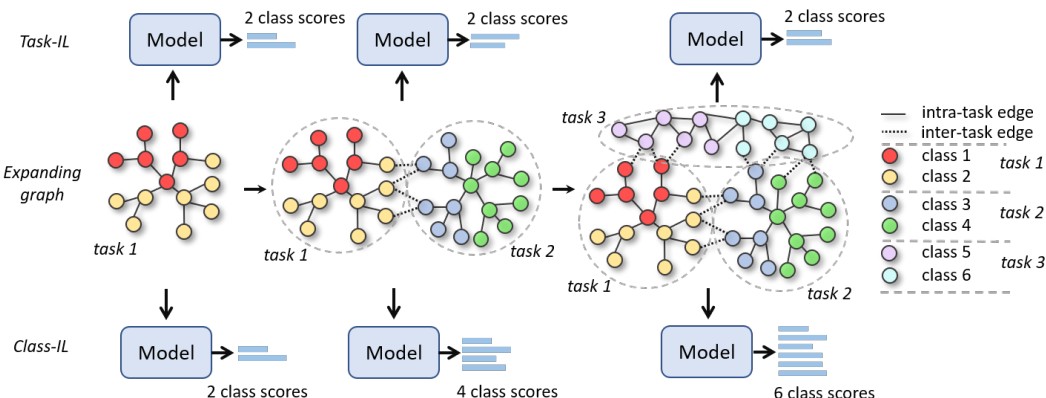

Figure 2: Illustration of the task-IL and class-IL learning scenario for N-CGL. From left to right, we show the process in which subgraphs (tasks) containing the new classes of nodes are incrementally attached to the existing graph. Different classes are denoted with different colors. The above part shows the task-IL learning scenario, in which the model only needs to distinguish the classes within the current task. The bottom part shows the class-IL scenario, in which the model has to distinguish all the classes from existing tasks.

## 3.1 Node-level CGL

Node-level CGL (N-CGL) focuses on the node classification problem on an expanding graph as shown in Figure 2. Formally, the learning process of N-CGL can be formulated as continually training a model on a sequence of subgraphs (tasks): $\mathcal{S} = \{\mathcal{G}_1, \mathcal{G}_2, ..., \mathcal{G}_T\}$. Each $\mathcal{G}_\tau$ is a newly emerging subgraph of the overall graph with new types of nodes. A $\mathcal{G}_\tau$ is associated with a node set $\mathbb{V}_\tau$ an edge set $\mathbb{E}_\tau$. The edges connecting the new subgraph to the overall graph are denoted as inter-task edges (dashed edges in Figure 2). For classification tasks, each node $v$ has a label $\mathbf{y}_v \in \{0,1\}^C$, where $C$ is the total number of classes.

For classical continual learning, when learning a new task, the model only has access to the data that is related to the current task. However, for N-CGL, attention has to be paid to the inter-task edges. If the data of existing tasks are strictly inaccessible, then the inter-task edges have to be removed. This is because the message passing mechanism of GNNs would inevitably aggregate information from previous tasks via the inter-task edges. However, a more practical setting is to keep the inter-task edges and only forbid access to the labels of existing tasks. In other words, the supervised learning objective is calculated based on the nodes of the current task, but the node features from the previous tasks are leveraged. In CGLB, we provide both protocols for a comprehensive comparison. We also experimentally explore the difference between these two protocols in Section 4.1. In practice, each new subgraph may contain both new types of nodes and old types of nodes. Learning under this scenario is much easier since the data from existing tasks can be leveraged to alleviate the forgetting. Therefore, we do not consider this scenario in this work.

As shown in Table 1, we construct four benchmark datasets for N-CGL based on four public data sources: OGB-Arxiv, OGB-Products, Reddit, and CoraFull. We use the suffix -CL (continual learning) to denote that the dataset is for both task-IL and class-IL scenarios. In the following, we also use -tIL or -cIL to denote datasets solely used for task-IL or class-IL setting. To construct these datasets, we first remove the extremely small classes. For example, in OGB-Products, we removed the $47$-$th$ class that only has one node and cannot be divided into training, validation, and testing sets. Then, given a dataset with $C$ classes, we split these classes into $\lfloor \frac{C}{K} \rfloor$ groups with a specified order, so that each group containing $K$ classes of nodes. The graph is divided into $K$ subgraphs accordingly. In continual learning, more tasks will result in more severe forgetting problem. This inspires us to set default $K$ as 2 for CGLB, which maximizes the number of tasks to test the limit of given CGL methods. It is also flexible to set $K$ as other integers by simply specifying the parameter when importing the dataset class in CGLB. As for the order of the classes, to facilitate the comparison across different works, the default order is set by CGLB as the original class order within the source dataset. The order can also be set by the users for specific explorations.

---

[1]https://ogb.stanford.edu/docs/nodeprop/#ogbn-arxiv

[2]https://ogb.stanford.edu/docs/nodeprop/#ogbn-products

Table 1: The detailed statistics of the constructed benchmark datasets for N-CGL.

| Benchmark datasets | CoraFull-CL | Arxiv-CL | Reddit-CL | Products-CL |
|---|---|---|---|---|
| Data source | CoraFull [34] | OGB-Arxiv[1] | Reddit [17] | OGB-Products[2] |
| Learning scenario | task-IL & class-IL | task-IL & class-IL | task-IL & class-IL | task-IL & class-IL |
| # nodes | 19,793 | 169,343 | 227,853 | 2,449,028 |
| # edges | 130,622 | 1,166,243 | 114,615,892 | 61,859,036 |
| # classes | 70 | 40 | 40 | 46 |
| # tasks | 35 | 20 | 20 | 23 |
| average # nodes per task | 660 | 8,467 | 11,393 | 122,451 |
| average # edges per task | 4,354 | 58,312 | 5,730,794 | 2,689,523 |

## 3.2 Graph-level CGL

The majority of graph-level representation learning tasks deal with molecule property prediction. For G-CGL, as shown in Table 2, our benchmark datasets are constructed from SIDER, PubChemBioAssayAromaticity, and Tox21. Among these three datasets, different from N-CGL where examples of data (nodes) are connected causing mutual influence and possible inter-task correlation (by inter-task edges) during learning, there is no inter-example influence in G-CGL since the data are individual graphs (*e.g.*, the individual molecule graphs). In the task-IL setting of G-CGL, different tasks correspond to different molecule properties and each task is a binary classification problem. Therefore, the graphs of different tasks in the task-IL setting of G-CGL may overlap, because the same graph can have different properties. Among these datasets, not all graphs are labeled with all properties. Therefore, for each task, the unlabelled graphs are removed. The detailed statistics of these graph datasets are shown in Table 2.

## 3.3 Task-incremental and class-incremental learning settings

Based on the fact that whether task indicators are provided to the model during testing, continual learning can be divided into task-incremental (task-IL) and class-incremental (class-IL) settings.

- In task-IL setting, a model requires the task indicator to perform inference on a given task, and the model only needs to distinguish different classes within each task without considering the classes from existing tasks.

- In class-IL setting, task indicators are not provided for the model and the model has to distinguish among all classes from the current and existing tasks.

Concretely, suppose the model has learned on a citation network with a sequence of two tasks {(*physics*, *chemistry*), (*biology*, *math*)}. In class-IL, a given document could come from any of these four classes, and the model is required to classify the given document into one of the four classes. However, in task-IL, a given document comes with a task indicator telling the model whether it is from (*physics*, *chemistry*) or (*biology*, *math*). Then, the model is only required to classify the given document into to (*physics*, *chemistry*) or (*biology*, *math*), without being able to distinguish between *physics* and *biology* or between *chemistry* and *math*. Obviously, for the commonly encountered application scenarios like the citation network or social networks in N-CGL where the tasks are multi-class classification, the class-IL is more practical for N-CGL but is also more challenging. However, for the multi-label classification, like the molecule property prediction tasks in G-CGL, the task-IL and class-IL are equivalent because the model has to give a binary prediction for each task let alone the task orders.

## 3.4 Evaluation metrics and visualization

Different from the traditional learning setting, which typically requires only a single numerical value to indicate the overall performance, CGL is trained and evaluated on multiple tasks and the evaluation protocols should reflect the learning dynamics. Therefore, the most thorough metric to evaluate

---

[3] `https://tripod.nih.gov/tox21/challenge/`

Table 2: The detailed statistics of the constructed benchmark datasets for G-CGL.

| Benchmark datasets | SIDER-tIL | Aromaticity-CL | Tox21-tIL |
|---|---|---|---|
| Data source | SIDER [53] | PubChemBioAssayAromaticity [54] | Tox21 [3] |
| Learning scenario | task-IL | task-IL & class-IL | task-IL |
| # graphs | 1,427 | 3,868 | 7,831 |
| # nodes | 48,006 | 115,061 | 145,459 |
| # edges | 100,912 | 253,018 | 302,190 |
| # classes | 27 | 30 | 12 |
| # tasks | 27 | 15 | 12 |
| average # graphs per task | 53 | 155 | 653 |
| average # nodes per task | 1,778 | 7,671 | 12,122 |
| average # edges per task | 3,737 | 16,868 | 25,183 |

a CGL model is the performance matrix $\mathrm{M}^p \in \mathbb{R}^{T \times T}$, where $\mathrm{M}^p_{i,j}$ denotes the performance (*e.g.* accuracy, AUC-ROC, *etc.*) on task $j$ after the model has been trained over a sequence of tasks from 1 to $i$.

To better understand the dynamics of the overall performance while learning on the task sequence, we also provide the toolkit to draw the learning dynamics curve, which is a sequence of the average performance (AP): $\left\{ \frac{\sum_{j=1}^{i} \mathrm{M}^p_{i,j}}{i} | i = 1, ..., T \right\}$ and a sequence of the average forgetting (AF): $\left\{ \frac{\sum_{j=1}^{i-1} \mathrm{M}^p_{i,j} - \mathrm{M}^p_{j,j}}{i-1} | i = 2, ..., T \right\}$ when the number of learned tasks varies. AP and AF were proposed by Lopez *et al.*[32] and widely adopted by following works on continual learning. Specifically, after learning over a sequence of tasks from 1 to $i$, the model's performance on each previous task is contained in the $i$-th row of $\mathrm{M}^p$ (*i.e.*, $\{\mathrm{M}^p_{i,j} | j = 1, ..., i\}$), therefore, the average over the $i$-th row denotes the average performance on all previous tasks when the model has finished learning from task 1 to $i$. As for the AF, with $\{\mathrm{M}^p_{i,j} - \mathrm{M}^p_{j,j} | j = 1, ..., i - 1\}$ denoting the performance decrease (forgetting) on task $j$ after learning task $i$, its average value denotes the average forgetting of each learnt task after learning from task 1 o task $i$. To use a single numerical value to quantify the overall learning performance, the AP and AF after learning all $T$ tasks can be used. In our library, the output is formatted in standard forms and can be directly evaluated with these metrics with the evaluation protocol with a one-line function.

## 4 Baseline Implementations and Result Analysis

In this section, we tested state-of-the-art continual learning (CL) methods on CGLB. These CL methods are implemented based on GNN backbones. A brief introduction of the implemented CL methods is as below:

1. **Bare model** denotes the backbone GNN without continual learning technique. Therefore, this can be viewed as the lower bound on the continual learning performance.

2. **Elastic Weight Consolidation (EWC) [25]** adds a quadratic penalty on the model weights according to their importance to the previous tasks to maintain its performance on existing tasks.

3. **Memory Aware Synapses (MAS) [2]** is also based on regularization. Different from EWC, MAS evaluates the parameter importance according to the sensitivity of the predictions on the parameters.

4. **Gradient Episodic Memory (GEM) [32]** stores representative data in the episodic memory. During learning, GEM modifies the gradients of the current task with the gradient calculated with the stored data to prevents the loss of the previous tasks from increasing.

5. **Topology-aware Weight Preserving (TWP) [31]** adds a penalty to preserve the topological information of the previous graphs.

Table 3: Performance comparisons under task-IL without inter-task edges on different datasets (↑ higher means better).

| C.L.T. | CoraFull-CL | | Arxiv-CL | | Reddit-CL | | Products-CL | |
|---|---|---|---|---|---|---|---|---|
| | AP/% ↑ | AF/% ↑ | AP/% ↑ | AF /% ↑ | AP/% ↑ | AF /% ↑ | AP/% ↑ | AF /% ↑ |
| Bare model | 58.0±1.7 | -38.4±1.8 | 61.7±3.8 | -28.2±3.3 | 73.6±3.5 | -26.9±3.5 | 67.6±1.6 | -25.4±1.6 |
| EWC [25] | 78.9±2.4 | -15.5±2.3 | 78.8±2.7 | -5.0±3.1 | 91.5±4.2 | -8.1±4.6 | 90.1±0.3 | -0.7±0.3 |
| MAS [2] | 93.0±0.5 | -0.6±0.2 | 88.4±0.2 | -0.0±0.0 | 98.6±0.5 | -0.1±0.1 | 91.2±0.6 | -0.5±0.2 |
| GEM [32] | 91.6±0.6 | -1.8±0.6 | 87.3±0.6 | 2.8±0.3 | 91.6±5.6 | -8.1±5.8 | 87.8±0.5 | -2.9±0.5 |
| TWP [31] | 92.2±0.5 | -0.9±0.3 | 86.0±0.8 | -2.8±0.8 | 87.4±3.8 | -12.6±4.0 | 90.3±0.1 | -0.5±0.1 |
| LwF [29] | 56.1±2.0 | -37.5±1.8 | 84.2±0.5 | -3.7±0.6 | 80.9±4.3 | -19.1±4.6 | 66.5±2.2 | -26.3±2.3 |
| ER-GNN [64] | 90.6±0.1 | -3.7±0.1 | 86.7±0.1 | 11.4±0.9 | 98.9±0.0 | -0.1±0.1 | 89.0±0.4 | -2.5±0.3 |
| Joint | 95.2±0.2 | - | 90.3±0.2 | - | 99.4±0.1 | - | 91.8±0.2 | - |

Table 4: Performance comparisons under task-IL on different datasets with inter-task edges (↑ higher means better).

| C.L.T. | CoraFull-CL | | Arxiv-CL | | Reddit-CL | | Products-CL | |
|---|---|---|---|---|---|---|---|---|
| | AP/% ↑ | AF/% ↑ | AP/% ↑ | AF /% ↑ | AP/% ↑ | AF /% ↑ | AP/% ↑ | AF /% ↑ |
| Bare model | 60.9±2.1 | -35.7±2.1 | 57.7±1.2 | -31.2±1.0 | 67.4±5.5 | -22.9±6.1 | 60.4±1.5 | -31.5±1.5 |
| EWC [25] | 73.9±2.6 | -21.9±2.6 | 67.6±3.2 | -17.8±3.0 | 92.6±2.2 | -6.8±2.4 | 73.6±1.3 | -25.2±2.3 |
| MAS [2] | 93.6±0.6 | -0.9±0.2 | 88.5±0.4 | 0.1±0.1 | 95.9±0.5 | -2.2±0.4 | 78.3±1.6 | -0.2±0.8 |
| GEM [32] | 91.8±0.7 | -3.1±0.6 | 78.7±1.2 | -4.5±0.6 | 77.1±11.1 | -23.2±11.7 | 80.6±0.9 | -9.0±1.0 |
| TWP [31] | 90.3±1.1 | -4.1±1.0 | 87.1±0.7 | -1.2±0.4 | 91.2±1.8 | -8.1±1.9 | 88.2±0.8 | -1.8±1.0 |
| LwF [29] | 61.3±2.3 | -34.9±2.3 | 83.8±1.1 | -5.8±1.1 | 81.8±2.7 | -18.1±2.8 | 71.4±2.0 | -26.1±1.9 |
| ER-GNN [64] | 90.9±0.7 | -4.8±0.6 | 87.2±0.2 | 11.8±0.6 | 97.0±0.3 | -2.2±0.3 | 88.9±0.2 | 0.6±0.8 |
| Joint | 95.4±0.2 | - | 89.4±0.3 | - | 98.7±0.1 | - | 87.6±0.5 | - |

6. **Learning without Forgetting (LwF) [29]** minimizes the discrepancy between the logits of the old model and the new model (knowledge distillation) to preserve knowledge from the old tasks.

7. **Experience Replay GNN (ER-GNN) [64]** integrates memory-replay to GNNs by storing representative nodes selected from previous tasks.

8. **Joint Training** does not follow the continual learning setting and trains the backbone GNN on all tasks simultaneously. Therefore, Joint Training has no forgetting problems and its performance can be viewed as the upper bound for continual learning.

In the paper, a 2-layer GCN [24] backbone is adopted for all CL methods for a fair comparison, while the results with other GNN backbones are discussed in Section 2 of the Appendix.

### 4.1 N-CGL under the task-IL scenario

In this subsection, we tested state-of-the-art methods on CGLB under the widely adopted task-IL scenario. We also studied the influence of the inter-task edges on the performance, which is currently ignored by the existing works. First, in Table 3, we show the final AP and AF (defined in Section 3.4) of each method after learning each entire task sequence. On average, the Bare model without any continual learning technique performs the worst, and Joint training performs the best. The AF is inapplicable to joint trained models because they do not follow the continual learning setting and are simultaneously trained on all tasks. In terms of both the final AP and AF, all methods significantly outperform the Bare model especially the regularization based ones like EWC, MAS, and TWP, which demonstrates that the regularization methods indeed can alleviate the forgetting problem.

In Table 4, we show the results of N-CGL under task-IL scenario with inter-task edges being kept. Comparing Table 3 and 4, the inter-task edges are indeed causing significant difference in the performance. However, according to the results, the difference cannot be simply summarized as one is more superior than the other, because the influence factors are multi-fold and the performance difference also varies across different datasets and methods. Specifically, multiple factors including both beneficial and harmful ones are governing the performance difference, which is analyzed below.

1. Each node is predicted based on its neighborhood by GNNs. The inter-task edges alter the neighborhoods of the boundary nodes (forming edges to new subgraphs) of the previous

Table 5: Performance comparisons under class-IL on different datasets without inter-task edges (↑ higher means better).

| C.L.T. | CoraFull-CL | | Arxiv-CL | | Reddit-CL | | Products-CL | |
|---|---|---|---|---|---|---|---|---|
| | AP/% ↑ | AF/% ↑ | AP/% ↑ | AF /% ↑ | AP/% ↑ | AF /% ↑ | AP/% ↑ | AF /% ↑ |
| Bare model | 2.9±0.0 | -94.7±0.1 | 4.9±0.0 | -87.0±1.5 | 5.1±0.3 | -94.5±2.5 | 3.4±0.8 | -82.5±0.8 |
| EWC [25] | 15.2±0.7 | -81.1±1.0 | 4.9±0.0 | -88.9±0.3 | 10.6±1.5 | -92.9±1.6 | 3.3±1.2 | -89.6±2.0 |
| MAS [2] | 12.3±3.8 | -83.7±4.1 | 4.9±0.0 | -86.8±0.6 | 13.1±2.6 | -35.2±3.5 | 15.0±2.1 | -66.3±1.5 |
| GEM [32] | 7.9±2.7 | -84.8±2.7 | 4.8±0.5 | -87.8±0.2 | 28.4±3.5 | -71.9±4.2 | 5.5±0.7 | -84.3±0.9 |
| TWP [31] | 20.9±3.8 | -73.3±4.1 | 4.9±0.0 | -89.0±0.4 | 13.5±2.6 | -89.7±2.7 | 3.0±0.7 | -89.7±1.0 |
| LwF [29] | 2.0±0.2 | -95.0±0.2 | 4.9±0.0 | -87.9±1.0 | 4.5±0.5 | -82.1±1.0 | 3.1±0.8 | -85.9±1.4 |
| ER-GNN [64] | 3.0±0.1 | -93.8±0.5 | 30.3±1.5 | -54.0±1.3 | 88.5±2.3 | -10.8±2.4 | 24.5±1.9 | -67.4±1.9 |
| Joint | 80.6±0.3 | - | 46.4±1.4 | - | 99.3±0.2 | - | 71.5±0.7 | - |

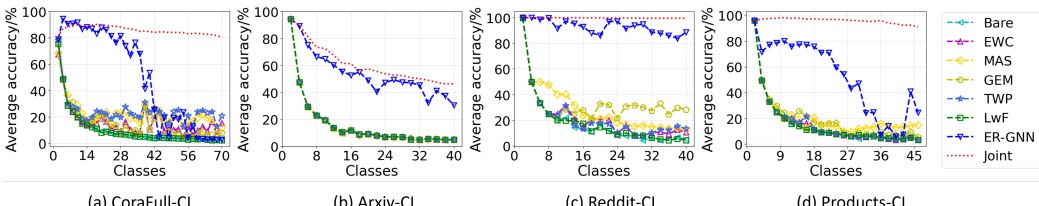

(a) CoraFull-CL     (b) Arxiv-CL     (c) Reddit-CL     (d) Products-CL

Figure 3: Dynamics of the AP during learning on the task sequences of different datasets.

    tasks, causing a concept drift. Since the training on these affected nodes is not repeated, this could harm the performance.

2. Due to the message passing mechanism of GNNs, the inter-task edges also break the restriction on the access to the information of the previous tasks. Real-world graphs often exhibit the 'small-world' property which means the shortest path between any two nodes is mostly short. Therefore, the inter-task edges are likely to propagate a significant amount of information from old tasks to new tasks. This effect may alleviate the forgetting problem and be beneficial for performance.

3. With inter-task edges, more information about the neighborhood patterns is provided for the boundary nodes. By examining Joint training, which is not influenced by the two effects mentioned above, we can find that the influence of inter-task edges highly depends on the datasets.

In a nutshell, the inter-task edges are introducing contradictory factors influencing the performance, and whether the net influence is beneficial or harmful highly depends on the properties of the datasets.

Another observation is that the performance of some methods exhibit dependencies on the datasets. For example, LwF performs well on Arxiv-CL, Reddit-CL, and Products-CL under both settings with and without inter-task edges, but performs similarly to Bare model on CoraFull-CL under both settings. Overall, existing methods could perform well in N-CGL under the task-IL scenario, and some are even close to Joint training. However, for N-CGL, it is more practical (yet challenging) to conduct a class-IL setting, as explained in Section 3.1, which is therefore studied in the next subsection.

## 4.2 N-CGL under the class-IL scenario

Different from the task-IL, class-IL is currently rarely studied for CGL. With the final AP and AF showed in Table 5, the first impression is that the performance is much worse compared to the task-IL scenario. This is not solely caused by the forgetting problem but also by the increased number of classes (difficulty) in each task. For example, on CoraFull-CL with 35 tasks and 2 classes per task, the task-IL scenario only requires the model to perform a 2-class classification for each task. While in class-IL, the final evaluation requires the model to perform a 70-class classification. Despite the difficulty in the increasing number of classes, forgetting is still the governing factor in the performance, since Joint training obtains pretty high performance. To further understand the learning dynamics with the task sequence, we show the learning dynamics curve and visualize the performance matrix as defined in Section 3.4. The learning curves, as shown in Figure 3, reveal the

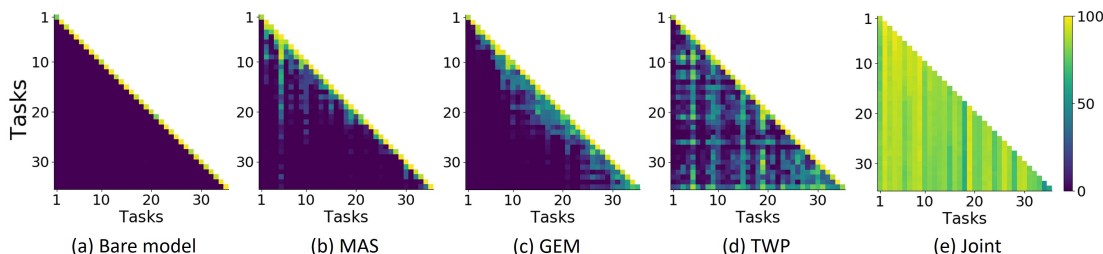

Figure 4: Visualization of the performance matrices of different continual learning methods on CoraFull-CL. Each entry in these matrices represent the performance on task $j$ (column) after learning task $i$ (row).

decreasing patterns of the AP for different methods. The long task sequence constructed in CGLB with tens of tasks is indeed challenging. Many methods with final AP close to the Bare model in Table 5 outperform the Bare model when fewer tasks are learned. The performance is also highly dependent on the datasets. On the most challenging Arxiv-CL, most methods experience strong forgetting problems, and even the Joint training obtains a much lower performance compared to other datasets. Since the memory-replay based method (ER-GNN) performs better than the regularization based techniques, this implies that the distribution discrepancies across different tasks of Arxiv-CL are strong and the regularization cannot simultaneously maintain the performance on previous tasks and keep the flexibility to adapt to new tasks. In contrast, different methods on CoraFull-CL and Reddit-CL exhibit reasonable improvement compared to the Bare model. Overall, under the class-IL scenario, memory replay based techniques (ER-GNN) appear to be more effective compared to other CGL techniques.

To gain a better understanding of the behaviors of different baselines, we visualize the performance matrices, the most thorough evaluation metric, of several representative baselines in Figure 4. We first observe two representative patterns of the Bare model and the Joint training. The Bare model has no problem learning each new task (the diagonal entries, better viewed with zooming in), but experiences a catastrophic forgetting of the previously learned tasks. In contrast, Joint training demonstrates a perfect performance that both learn well on new tasks and exhibit no forgetting on each existing task (each column denotes the performance change of one task along with the learning on the task sequence). The baselines, *i.e.*, MAS, GEM, and TWP, lie somewhere between these two extreme cases. Since they all have certain constraints on the changes of the model parameters, the ability to adapt to new tasks is weaker than the Bare model (*i.e.* the diagonal entries have smaller values). But the regularization also enables the model to maintain its performance over existing tasks to a certain extent. By observing each column, we could see that the performance of each task gradually decreases along the learning process, while the Bare model experiences an abrupt decrease on the latest existing task immediately after learning the new task. Due to space limitations, the results of other techniques are provided in Section 2 of the Appendix.

## 4.3   G-CGL

In this subsection, we study the G-CGL for both task-IL and class-IL scenarios. The performance matrix contains the AUC-ROC of each task for the task-IL setting, and the accuracy of each task for the class-IL setting. As shown in Table 6, the task-IL is still relatively easier and all baseline methods gain reasonable improvement against the bare model. In contrast, all baseline methods perform badly under the class-IL scenario. Since the Joint training obtains satisfying performance, it is clear that the failures of the baselines are not caused by the backbone GNNs but are caused by the forgetting problem. Different tasks of G-CGL correspond to different topology distributions of graphs. The topology distributions appear to be highly different, which is also verified by the results that GEM often encounters the no solution error. Essentially, GEM is developed based on clipping the gradients of the current task so that the update directions do not lead to the loss increase of the previous tasks. Therefore, the no solution error implies that the task distributions are so different that learning some of the new tasks will inevitably increase the loss of some previous tasks. This error is not encountered in N-CGL, which implies that the distribution discrepancy is more severe in G-CGL. More detailed studies to further justify the above analysis could be obtained with the learning dynamics curves and the performance matrices, which are included in Section 2 of the Appendix due to space limitations.

Table 6: Performance comparisons on different graph-level prediction datasets (↑ higher means better).

| C.L.T. | SIDER-tIL task-IL | | Tox21-tIL task-IL | | Aromaticity-CL task-IL | | Aromaticity-CL class-IL | |
|---|---|---|---|---|---|---|---|---|
| | AP ↑ | AF ↑ | AP ↑ | AF ↑ | AP/% ↑ | AF /% ↑ | AP/% ↑ | AF /% ↑ |
| Bare model | 0.577±0.021 | -0.070±0.023 | 0.717±0.012 | -0.138±0.015 | 53.2±3.8 | -42.4±7.5 | 5.4±0.2 | -66.3±2.0 |
| EWC [25] | 0.604±0.001 | -0.012±0.009 | 0.811±0.005 | -0.015±0.002 | 58.0±5.0 | -22.8±2.5 | 7.5±1.6 | -69.9±2.5 |
| MAS [2] | 0.627±0.018 | 0.001±0.001 | 0.798±0.009 | -0.022±0.004 | 63.8±1.3 | -24.0±2.3 | 7.8±1.0 | -80.5±2.1 |
| GEM [32] | 0.655±0.010 | 0.037±0.002 | 0.825±0.008 | 0.024±0.012 | 75.9±11.2 | 12.2±7.7 | 9.3±2.4 | -75.6±1.1 |
| TWP [31] | 0.611±0.015 | -0.005±0.002 | 0.720±0.024 | -0.037±0.018 | 54.5±1.3 | -21.4±0.2 | 6.5±1.6 | -45.4±7.7 |
| LwF [29] | 0.610±0.021 | -0.056±0.035 | 0.835±0.007 | -0.019±0.006 | 58.8±3.9 | -28.7±5.5 | 5.7±3.3 | -16.1±4.7 |
| Joint | 0.692±0.068 | - | 0.826±0.005 | - | 75.4±6.0 | - | 77.2±1.3 | - |

## 5   Conclusion

In this paper, we systematically analyzed the setup of continual graph learning and proposed the CGLB to thoroughly and fairly assess state-of-the-art methods from different perspectives, *i.e.*, N-CGL and G-CGL under both task-IL and class-IL scenarios. Besides, we also tested several representative existing techniques on CGLB with detailed analysis with the model performance. By providing the CGLB as well as the toolkit for training, evaluation, and visualization, we greatly lower the barrier to explore CGL problem. In the future, we will also keep adopting new datasets and new benchmark results into CGLB. We will also pursue more practical settings for CGL.

## 6   Future Works & Maintenance Plan

Continual graph learning is a newly emerging area, and inevitably more datasets, benchmarks, and evaluation tools will be needed to facilitate its development. Therefore, we will keep adding new contents to CGLB including new benchmark tasks constructed from other data sources, new benchmark results obtained from new methods, as well as an improved toolkit for the possible new evaluation requirements. All materials are accessible through `https://github.com/QueuQ/CGLB`.

Currently, CGLB only includes homogeneous graph datasets for NCGL tasks. In the future, we will also include benchmarks on heterogeneous graphs. First, we are now constructing benchmark tasks from several widely adopted data sources for heterogeneous graphs, including DBLP (academic network), IMDB (film rating network), Yelp (social media network), Amazon (E-commercial network). IMDB and Amazon are obtained via `https://grouplens.org/datasets/movielens/100k/` and `http://jmcauley.ucsd.edu/data/amazon/`. While the curation of DBLP and Yelp follows the process proposed in a heterogeneous graph benchmark work [56]. The task construction will follow the node-level continual graph learning (NCGL) task construction proposed in our paper, with both task-IL and class-IL scenarios. Second, to implement the continual learning baselines on heterogeneous graphs, heterogeneous GNNs will have to be implemented as the backbones. Several representative heterogeneous GNNs including Heterogeneous Graph Neural Network (HetGNN) [62], Heterogeneous Graph Attention Network (HAN) [49], Heterogeneous Graph Transformer (HGT) [22], Relational Graph Convolutional Networks (R-GCNs) [41], Metapath2vec [11], and Predictive Text Embedding (PTE) [44] will be implemented into our continual learning pipelines, which will be benchmarked with the constructed tasks.

We will also keep exploring new learning scenarios. For example, continual learning on evolving graphs, a straightforward extension of our current work, is one of our future works. Specifically, an evolving graph may experience constant changes in both nodes and edges over time, and a model that constantly adapts to the newly incoming patterns may also experience forgetting the patterns learned from past periods. Therefore, the observational data of an evolving graph may be split into multiple periods, each of which corresponds to one task. Moreover, when the temporal boundaries are not clear, the setting will become task-free [3]. Real-world data exhibiting such behaviors include the financial data, traffic data, *etc.*. Besides, domain-incremental learning (domain-IL) [45, 63] can also be directly obtained by replacing the task-specific output heads with one head shared by all tasks and will be included in our benchmark. Specifically, an important aspect of constructing domain-IL tasks is to find the proper datasets in which data from different domains share the same label space. For example, proteins from different human tissues [17] or different species [21] can be viewed as data from different domains, but their functions (labels) are in the same space.

As mentioned above, all the potential new benchmark tasks and baselines, as well as any improvement or correction, will be updated in our repository. We also welcome any comments on improving our CGLB.

## 7 Acknowledgment

Dacheng Tao and Xikun Zhang are supported by Australian Research Council Projects FL-170100117, DP-180103424, IH-180100002, and IC-190100031. Dongjin Song is supported by UConn Scholarship Facilitation Fund (SFF) Award, NSF CNS-2154191, and USDA NIFA 2022-67022-36603.

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
