# OpenReview forum: "CGLB: Benchmark Tasks for Continual Graph Learning"
_NeurIPS.cc/2022/Track/Datasets_and_Benchmarks — NeurIPS 2022 Datasets and Benchmarks _

### Official Review · Reviewer_KuaG · 2022-07-22
**A comprehensive benchmark and a useful toolkit for Continual Graph Learning**

**Rating:** 7
**Confidence:** 4
**Correctness:** The evaluation methods and experiment…
**Clarity:** The paper is well written and easy to…

**Strengths:**

Significance: the contribution is significant as it lowers the barrier to exploring the CGL problem to a large extent. The constructed standard datasets, experimental setting and evaluation toolkit are beneficial for the development of CGL.

Relevance: the benchmark is relevant to the continual graph learning community.

Accessibility and Accountability: the documentation is mostly clear.


**Weaknesses:**

One concern is that the results of some existing SOTAs across different datasets is inconsistent. For example, Tables 3 and 4 show that NAS performs significantly better than LwF on the CoraFull-CL and Arxiv-CL datasets, whereas the opposite is true on the Reddit-CL and Products-CL datasets. Some detailed analysis is necessary.

**Additional Feedback:**

Please refer to the weakness above.

**Documentation:**

The authors provide sufficient and detailed instructions on Github.

**Ethics:**

No dedicated ethics review appears to be necessary.

**Relation To Prior Work:**

The discussion on prior work is clear and sufficient.

**Summary And Contributions:**

The authors present a number of contributions related to the CGL:
- a systematic taxonomy of different CGL scenarios that include node-level CGL and graph-level CGL;
- the differences between the task-incremental learning setting and the class-incremental learning setting;
- a useful toolkit for training, evaluation, and visualization, with a maintenance plan;
- a comprehensive study of state-of-the-art methods.

---

> ### Author Response · Authors · 2022-08-11
> **Responses to Reviewer KuaG**
>
> We sincerely thank the reviewer for the recognition of our work and the constructive comments. Detailed responses are provided below.
>
>
> **Q1. One concern is that the results of some existing SOTAs across different datasets is inconsistent. For example, Tables 3 and 4 show that MAS performs significantly better than LwF on the CoraFull-CL and Arxiv-CL datasets, whereas the opposite is true on the Reddit-CL and Products-CL datasets. Some detailed analysis is necessary.**
>
> **A1.** Thanks for raising this question. A method's performance over a dataset is determined by the characteristics of the dataset (e.g., data distribution, graph statistics, noises in data and labels, etc.).
>
> In the example mentioned by the reviewer, MAS outperforms LwF on CoraFull-CL and Arxiv-CL, which are both citation networks. While on Reddit-CL and Products-CL, which are social network and co-purchasing network, LwF outperforms MAS. MAS reduces the forgetting issue based on regularization on the model parameters, which also restricts the model's plasticity when adapting to new tasks. In contrast, LwF is based on knowledge distillation, which does not have constraints on the parameters and has more plasticity when learning new tasks. With the information above, a possible reason of the phenomenon mentioned by the reviewer is that the distribution similarity across different tasks in citation networks (CoraFull-CL and Arxiv-CL) is higher than that in Reddit-CL and Products-CL. A higher similarity in the distributions of different tasks requires less plasticity from the model for adapting to new tasks, and MAS may succeed. When the similarity is lower, more plasticity is required when learning new tasks and LwF may succeed.
>
> Please let us know if there is any remaining concern, and we are more than happy to address them.

---

> > ### Comment · Reviewer_KuaG · 2022-08-23
> > **Thanks for the response**
> >
> > The response has addressed my concern adequately and I will keep the current rating. Please update the discussions in the paper.

---

> > > ### Author Response · Authors · 2022-08-23
> > > **Thanks for the recognition of our work**
> > >
> > > We sincerely thank the reviewer for recognizing our contribution, and we are more than happy to know that the concerns were successfully resolved.
> > >
> > > The discussion has been updated in the paper in Section 4.2 (line 331-343).

---

### Official Review · Reviewer_qpB2 · 2022-07-22

**Rating:** 6
**Confidence:** 3

**Strengths:**

1. This paper defines the two tasks (node-level and graph-level continual learning), and also two learning settings (task/class incremental learning).
2. The experimental results show that there is some room for improvement.
3. The documentation in the Github repo looks good in general.

**Weaknesses:**

1. The rationale behind the designed metrics Average Performance (AP) and Average Forgetting (AF) are not provided. For AP, why not take an average over the entire matrix?
2. It looks like the performance matrix $M^p$ only considers sequences of two tasks, rather than a sequence of $T$ tasks stated in the section 3.1.
3. Some details are not documented well. For example,
- Is the order of different tasks fixed?
- What does $M_{i,i}^p$ mean?
4. This paper does not provide statistics for each task. For example, the average number of nodes/edges for each task; maybe also the difficulty level of different tasks. It will be better if the information can be included.


**Additional Feedback:**

No.

**Clarity:**

No, the rationale of the design for the performance matrix and evaluation metrics are not provided.

**Correctness:**

Not sure about the design of the performance matrix and evaluation metrics. More details should be provided.

**Documentation:**

Good in general. However, it is unclear that (1) is the order of tasks fixed? (2) the difficulty level of each class.

**Ethics:**

No.

**Relation To Prior Work:**

Yes.

**Summary And Contributions:**

This paper introduces a new benchmark dataset called Continual Graph Learning Benchmark (CGLB). The paper first builds a systematic taxonomy of different CGL scenarios including node-level CGL (N-CGL) and graph-level CGL (G-CGL). Then it further introduces two learning settings: task incremental learning (task-IL) and class incremental learning (class-IL). Next, the paper introduces how the dataset was built and also proposes some evaluation metrics to evaluate algorithms. Finally, several empirical studies are provided.

---

> ### Author Response · Authors · 2022-08-11
> **Responses to Reviewer qpB2 (Part 2)**
>
> (Citations below are for Q1)
>
> [1] Lopez-Paz, David, and Marc'Aurelio Ranzato. "Gradient episodic memory for continual learning." Advances in neural information processing systems 30 (2017).
>
> [2] Liu, Huihui, Yiding Yang, and Xinchao Wang. "Overcoming catastrophic forgetting in graph neural networks." Proceedings of the AAAI Conference on Artificial Intelligence. Vol. 35. No. 10. 2021.
>
> [3] Zhang, Xikun, Dongjin Song, and Dacheng Tao. "Hierarchical Prototype Networks for Continual Graph Representation Learning." IEEE Transactions on Pattern Analysis and Machine Intelligence (2022).
>
> **Q2. It looks like the performance matrix $M^p$ only considers sequences of two tasks, rather than a sequence of $T$ tasks stated in the section 3.1.**
>
> **A2.** Thanks for raising this question. $M^p$ actually considers a sequence of $T$ tasks (the detailed numbers of tasks $T$ are provided in Table 1 and 2) instead of only sequences of two tasks. Specifically, $M^p\in \mathbb{R}^{T\times T}$, where $T$ is the length of the task sequence.
> **Each entry $M^p_{i,j}$ denotes the performance on the $j$-th task after the model has been trained over a sequence of tasks from 1 to the $i$ (rather than just training the model only on task $i$ and getting the performance on task $j$).** We have added additional explanation on this to avoid confusion (line 208-209)
>
> Therefore, the $j$-th column of the matrix (i.e., {$M^p_{i,j}$, $i=j,...,T$}, $i$ doesn't start from 1 because $M^p$ is lower triangular) denotes how the model's performance on task $j$ varies after learning each new task $i$ ($i=j,...,T$).
> And the last ($T$-th) row of the matrix contains the model's performance on every learnt task after learning from task 1 to task $T$.
>
> In our experiments, we also visualized the performance matrix on task sequences of tens of tasks (Figure 4 in the paper, and Figure 1-5 in Appendix).
>
>
> **Q3. Some details are not documented well. For example, Is the order of different tasks fixed? What does $M^p_{i,i}$ mean?**
>
> **A3.** Yes, the order is fixed for a fair comparison among different methods (line 168-170 of the paper).
>
> $M^p_{i,i}$ denotes the model's performance on the task $i$ after the model has just been trained over a sequence of tasks from 1 to $i$ (line 207-209 of the paper).
>
>
> **Q4. This paper does not provide statistics for each task. For example, the average number of nodes/edges for each task; maybe also the difficulty level of different tasks. It will be better if the information can be included.**
>
> **A4.** Thanks for this suggestion, we have now added the corresponding information in the paper (Table 1 and 2). Specifically, for the NCGL datasets, we provide both the average number of nodes and edges per task (Table 1). For the GCGL datasets, we provide not only the average number of nodes and edges, but also the average number of graphs per task (Table 2). Besides, to provide more detailed statistics for each task, we have also added the number of nodes/edges/graphs in each class of the tasks for all datasets. And the results are shown in Section 6 (Table 5,6,7,8,9,10,11) of Appendix in the revised version. Since the tables are too long to be shown here in the response, please refer to our revised manuscript for these statistics.
>
> As for the difficulty level of each task, besides the task statistics, it is determined by multiple factors (e.g., the noise in both the data and the labels, the compatibility of the model and the data, etc.). In other words, the difficulty level of a task (to a specific model) is best shown by the model's performance after it has just learnt over this task. This corresponds to the diagonal entries of the performance matrices, which are shown in our experiments (Figure 4 in paper, and Figure 1-5 in Appendix).
>
> Please let us know if there is any remaining concern, and we are more than happy to address them.

---

> ### Author Response · Authors · 2022-08-11
> **Responses to Reviewer qpB2 (Part 1)**
>
> We sincerely thank the reviewer for raising these concerns. Detailed responses are provided below.
>
> **Q1. The rationale behind the designed metrics Average Performance (AP) and Average Forgetting (AF) are not provided. For AP, why not take an average over the entire matrix?**
>
> **A1.** Thanks for pointing out this confusion. Both Average Performance (AP) and Average Forgetting (AF) are well established metrics adopted by the existing continual graph learning and continual learning works (e.g., [1,2,3]).
>
> The rationale behind these two metrics is to measure a model's average performance/forgetting on all learnt tasks after learning a sequence of tasks. In the following, we will separately explain AP and AF with details and examples. We have also enriched the explanations in the paper for clarification (line 213-220).
>
> As we mentioned in Section 3.4 (line 207-208), in a performance matrix $M^p$, an entry $M^p\_{i,j}$ denotes the model performance on a learnt task $j$ ($j\leqslant i$) after learning the $i$-th task (i.e., the model has been trained over the sequence of tasks from 1 to $i$). Accordingly, the $i$-th row of the matrix $M^p$, i.e. {$M^p_{i,j}, j=1,...,i$} reflects the performance on each learnt task $j$ ($j=1,...,i$) after learning the $i$-th task.
>
> Therefore, the average performance (AP) after learning the $i$-th task is defined as the average of the $i$-th row of the matrix ({$M^p_{i,j}, j=1,...,i$}), because it reflects the model's average performance over all learnt tasks after learning a sequence of tasks (i.e., from the $1$-st task to the $i$-th task). As we could see, an AP can be computed after learning each new task, and the sequence of APs denotes how the overall performance varies with new tasks constantly coming in (i.e., the learning dynamics curves shown in Figure 3 of the paper).
>
> When a single numerical value is required to denote the model's overall performance (e.g., the results shown in Table 3, 4, and 5), as we mentioned in Section 3.4 (line 220-221), the AP computed after learning the entire sequence of tasks (i.e., $\frac{\sum_{j=1}^T \mathrm{M}^{p}_{T,j}}{T}$ computed over the last row of the performance matrix, {$M^p\_{T,j}, j=1,...,T$}) is used, which is the average model performance over all learnt tasks after the model has been trained over the sequence from the 1-st task to the final task. A high AP denotes that the model's overall performance on previous tasks is good after learning all the tasks sequentially (i.e., little forgetting issue). While a lower AP denotes that the model has more severe forgetting problem.
>
> The average forgetting (AF) is supported by the same rationale. Specifically, a diagonal entry $M^p\_{j,j}$ denotes the performance of a model when it has just learnt the task $j$ (i.e., before the performance on task $j$ is degraded because of the forgetting issue). The model will then keep learning the following tasks $j+1$, $j+2$, ..., etc.. After learning each following new task, the model is tested again on task $j$, and the performance on task $j$ becomes $M^p\_{j+1,j}$, $M^p\_{j+2,j}$, ..., etc.. At a specific step $i$, when the model has just learnt task $i$ ($i>j$), the performance on task $j$ becomes $M^p\_{i,j}$. Due to the forgetting issue (learning new tasks may interfere with the performance on task $j$), $M^p\_{i,j}$ may be lower than $M^p\_{j,j}$, and $M^p_{i,j}-M^p_{j,j}$ can quantitatively measure the forgetting. Accordingly, negative $M^p\_{i,j}-M^p\_{j,j}$ denotes that the performance on task $j$ is negatively affected  (forgetting on task $j$) by the following tasks from $j+1$ to $i$, and larger $|M^p_{j+1,j}-M^p_{j,j}|$ denotes more severe forgetting. It is also possible that $M^p\_{j+1,j}-M^p\_{j,j}$ is positive, denoting that the learning on the following tasks has a positive influence on task $j$, which is rare according to the experimental results. Similar to AP, after learning each new task, we could compute an AF over all learnt tasks, and the sequence of AFs reflects the learning dynamics from the forgetting perspective. To use a single numerical value to denote the average forgetting over all learnt tasks after learning the final task $T$, as mentioned in Section 3.4 (line 220-221), we could use the AF computed after learning the $T$-th task, i.e. $\frac{\sum_{j=1}^{T-1} \mathrm{M}^p_{T,j}-\mathrm{M}^p_{j,j}}{T-1}$ (reported in Table 3,4, and 5).
>
> With the rationale behind the AP and AF, the problem of directly taking an average over the entire performance matrix is straightforward. After learning $T$ tasks, only the last row (the $T$-th row) of the matrix reflects the performance of the current model on all tasks, while the other rows are not up-to-date information. Therefore, the average over the entire matrix would incorporate information that does not reflect the current up-to-date model.

---

> ### Author Response · Authors · 2022-08-26
> **We are happy to address any remaining/further concern/question**
>
> We sincerely thank the reviewer for the questions and suggestions in the reviews. The questions on the evaluation metrics and the training details, as well as the suggestion on including the task statistics, are all very constructive. We have carefully revised our manuscript to further clarify these questions, and have added detailed task statistics for all datasets.
>
> Given the limited time for discussion, we would really appreciate if the reviewer could let us know if the concerns are resolved, and if there is any remaining/additional concern/question. We are more than happy to address any remaining/further concern/question.

---

> ### Author Response · Authors · 2022-08-28
> **A friendly reminder to Reviewer qpB2 that the discussion ends in one day**
>
> We sincerely thank the reviewer for the constructive reviews, and we have made great efforts to address all these concerns.
>
> Given that the discussion is ending in only one day, we would really appreciate if the reviewer could let us know if the concerns are resolved, and if there is any remaining/additional concern/question. We are more than happy to address any remaining/further concern/question.

---

> > ### Comment · Reviewer_qpB2 · 2022-08-29
> > **Thanks for the comments.**
> >
> > Thanks for the comments.
> >
> > Hope the authors can explain more about the performance matrix in the revised version, otherwise, it is quite confusing.

---

> > > ### Author Response · Authors · 2022-08-29
> > > **Thanks for the recognition of our work and the constructive suggestion**
> > >
> > > We sincerely thank the reviewer for the recognition of our contribution and the constructive suggestion to enrich the explanations on the performance matrix.
> > >
> > > Following the suggestion, we have now updated our manuscript. In the revised version, more detailed explanations on the evaluation metrics including the performance matrix are added as a new section in Appendix (Section 7).

---

### Official Review · Reviewer_ynGE · 2022-07-26
**Could be a good benchmark paper by adding new method and more insights**

**Rating:** 6
**Confidence:** 5
**Clarity:** Yes, this paper is well written and e…

**Strengths:**

1. Graph continual learning is largely underexplored and requires a benchmark to promote future research.

2. This paper formulates different settings and also have the corresponding evaluations.

3. The paper is overall well-written and easy-to-follow.

**Weaknesses:**

1. The training label ratio is relatively large (60% for node-level evaluation and 80% for graph-level evaluation). However, this is not practical since new tasks or new classes only contain fewer labeled data than existing ones. It would be better to show the evaluation results with less training labels in the experiments.

2. The results seem quite random in task incremental setting (with inter-edges or without inter-edges), though the authors provide some possible explanations, more theoretical analysis is needed to give deeper understanding for this problem.

2. It would be much better if the authors could propose a simple and effective method after having the analysis on the existing methods. Based on the experiment results, what are the most important factor for addressing continual graph learning, especially for the class incremental setting? New insights for promoting the future research is somehow missing in the paper.




**Additional Feedback:**

Please refer to the weaknesses.

**Correctness:**

The claims in the paper are overall correct. The experiment design is reasonable and comprehensive.

**Documentation:**

The authors include a repository to support reproducibility in this work.

**Relation To Prior Work:**

Yes, this is the first benchmark paper in continual graph learning.

**Summary And Contributions:**

This paper is a benchmark paper that focuses on both node-level and graph-level graph continual learning. In particular, this paper studies two settings, i.e., task-incremental setting and class-incremental setting. Based on the established taxonomy, the authors construct the Continual Graph Learning Benchmark and evaluate a list of baselines under different evaluation settings. This paper also provides the toolkit for training, evaluation, and visualization, which can better promote the future research in this field.

---

> ### Author Response · Authors · 2022-08-11
> **Responses to Reviewer ynGE (Part 1)**
>
> We sincerely thank the reviewer for the recognition of our work and the constructive comments. Detailed responses are provided below.
>
> **Q1. The training label ratio is relatively large (60\% for node-level evaluation and 80\% for graph-level evaluation). However, this is not practical since new tasks or new classes only contain fewer labeled data than existing ones. It would be better to show the evaluation results with less training labels in the experiments.**
>
> **A1.** Thanks for this suggestion. In the revised version, we further set the splitting as 20\% for training, 40\% for validation and 40\% for test, and added the experimental results of both node-level and graph-level tasks in Appendix (Section 2.4). The results and analysis are also shown below for convenience. In our proposed CGLB, the train-validation-test splitting can be conveniently modified by simply specifying the corresponding arguments in our implemented pipelines.
> The corresponding explanations on specifying the training label ratio are also updated in both our GitHub repository (https://github.com/QueuQ/CGLB#modifying-the-train-validation-test-splitting) and our recently constructed documentation webpage (https://cglb--84.org.readthedocs.build/en/84/usage.html#pipeline-usages).
>
>
> Table 1: Performance comparisons under task-IL without inter-task edges on different node-level datasets with the splitting of train (20\%), validation (40\%), test (40\%).
>
> |C.L.T. |CoraFull-CL| |Arxiv-CL| |Reddit-CL| |Products-CL| |
> |---------| ------- |-|-------|-|---------|-|-------|-|
> | |AP/\% $\uparrow$ | AF/\% $\uparrow$ | AP/\% $\uparrow$ | AF/\% $\uparrow$ |AP/\% $\uparrow$ |AF/\% $\uparrow$|AP/\% $\uparrow$ |AF/\% $\uparrow$|
> |Bare model | 53.2$\pm$1.2|-40.5$\pm$1.5 | 60.9$\pm$6.2|-27.8$\pm$5.9 | 80.0$\pm$7.4|-20.2$\pm$7.7 | 66.0$\pm$1.9|-26.8$\pm$2.1|
> |EWC | 70.1$\pm$3.7|-20.9$\pm$4.1 | 68.4$\pm$4.5|-16.6$\pm$5.0 | 92.5$\pm$6.6|-7.1$\pm$6.9 | 90.3$\pm$0.7|-0.6$\pm$0.3|
> |MAS | 90.1$\pm$1.1|-0.5$\pm$0.5 | 87.3$\pm$0.2|0.0$\pm$0.0 | 98.9$\pm$0.3|0.0$\pm$0.0 | 91.8$\pm$0.7|-0.5$\pm$0.1
> |GEM | 90.0$\pm$0.1|0.0$\pm$0.4 | 80.4$\pm$0.1|-4.0$\pm$0.3 | 99.3$\pm$0.0|0.0$\pm$0.1 | 87.6$\pm$0.9|-3.0$\pm$0.8|
> |TWP | 86.9$\pm$0.5|-2.0$\pm$0.5 | 86.1$\pm$0.8|-1.7$\pm$0.7 | 91.2$\pm$3.6|-8.5$\pm$3.8 | 90.3$\pm$0.3|-0.6$\pm$0.3|
> |LwF | 54.2$\pm$1.6|-39.7$\pm$1.6 | 68.6$\pm$4.9|-20.6$\pm$5.3 | 78.7$\pm$4.3|-21.7$\pm$4.6 | 66.6$\pm$1.8|-26.9$\pm$2.0|
> |ER-GNN | 83.6$\pm$0.4|-6.7$\pm$0.5 | 89.2$\pm$0.1|5.6$\pm$0.5 | 98.8$\pm$0.1|-0.4$\pm$0.1 |89.3$\pm$0.1|-2.4$\pm$0.2|
> |Joint| 91.9$\pm$0.5|-  | 88.9$\pm$0.4|- | 99.4$\pm$0.0|-  | 91.2$\pm$0.8|-|
>
> Table 2: Performance comparisons on different graph-level prediction datasets with the splitting of train (20\%), validation (40\%), test (40\%).
>
> |C.L.T. | SIDER-tIL | |Tox21-tIL| | Aromaticity-CL | |Aromaticity-CL| |
> | ---------| ------- |-|-------|-|---------|-|-------|-|
> | | task-IL | | task-IL | | task-IL | | class-IL| |
> | |AP $\uparrow$ | AF $\uparrow$ | AP $\uparrow$ | AF $\uparrow$ |AP/\% $\uparrow$ |AF/\% $\uparrow$|AP/\% $\uparrow$ |AF/\% $\uparrow$|
> |Bare model | 0.532$\pm$0.007|0.028$\pm$0.016  | 0.645$\pm$0.022|0.119$\pm$0.015  | 52.0$\pm$1.4|0.1$\pm$1.2  | 4.3$\pm$1.4|-5.2$\pm$2.9 |
> |EWC| 0.503$\pm$0.012|0.003$\pm$0.006  | 0.602$\pm$0.021|0.028$\pm$0.027  | 52.0$\pm$1.4|0.1$\pm$1.2  | 3.9$\pm$0.8|-10.0$\pm$2.3  |
> |MAS| 0.518$\pm$0.010|0.014$\pm$0.006  |  0.630$\pm$0.022|0.092$\pm$0.029 | 58.8$\pm$2.0|5.0$\pm$2.2  | 3.8$\pm$1.0|-8.8$\pm$2.8 |
> |GEM| 0.578$\pm$0.006|0.072$\pm$0.014  |  0.685$\pm$0.007|0.183$\pm$0.025  | 70.6$\pm$2.2|18.8$\pm$3.4  | 10.9$\pm$1.7|2.1$\pm$3.5|
> |TWP| 0.505$\pm$0.009|0.009$\pm$0.004  |  0.593$\pm$0.010|0.046$\pm$0.025  | 54.2$\pm$2.7|0.9$\pm$2.9  | 3.5$\pm$1.1|-8.7$\pm$2.6|
> |LwF| 0.531$\pm$0.009|0.027$\pm$0.008  | 0.641$\pm$0.017|0.105$\pm$0.049  | 58.7$\pm$1.1|8.2$\pm$2.4  | 5.4$\pm$0.4|-8.4$\pm$0.9|
> |Joint | 0.575$\pm$0.009|-  | 0.678$\pm$0.017|-   | 69.4$\pm$1.0|-   | 35.4$\pm$3.9|- |
>
> Comparing Table 1 and 2 to Table 3 and 4 in paper, with less training data, the performance change of NCGL tasks (from Table 3 in paper to Table 1 here) and GCGL tasks (from Table 4 in paper to Table 2 here) are different. On NCGL tasks, with less training data, most methods exhibit performance decrease, but some methods perform better. While on GCGL tasks, almost all methods experience significant performance decrease. The reasons are two-fold. On the one hand, less training data may decrease the performance on single tasks. On the other hand, with less training data, the models adapt less to the new tasks, therefore the forgetting on previous tasks is less severe, which benefits the overall performance (AP and AF). The NCGL task sequences are longer than GCGL, and longer sequences bring more severe forgetting. Accordingly, the mitigation of the forgetting problem (by less training data) benefits the performance more in NCGL than GCGL. Therefore, the performance decreases more in GCGL than NCGL.

---

> > ### Author Response · Authors · 2022-08-24
> > **Responses to Reviewer ynGE (Part 2)**
> >
> > **Q2. The results seem quite random in task incremental setting (with inter-edges or without inter-edges), though the authors provide some possible explanations, more theoretical analysis is needed to give deeper understanding for this problem.**
> >
> > **A2.** Thanks for this suggestion. We are currently working on developing a theoretical framework to quantitatively understand the influence of the inter-task edges on the performance. It concerns multiple factors from both the dataset and the model structure perspectives. Here we provide our high-level ideas for the theoretical analysis.
> >
> > Specifically, based on the three factors analyzed in the paper (Section 4.1, line 275-288), our theoretical analysis focuses on the following perspectives.
> >
> > * How to quantitatively evaluate the concept drift on the boundary nodes and its effect on the performance, as well as their relationship.
> >
> > * How would the information aggregated from previous tasks (via the inter-task edges) quantitatively influence the performance.
> >
> > * How would the additional information aggregated via the inter-task edges quantitatively affect the performance on the boundary nodes.
> >
> > Finally, since these factors may counteract each other, a framework is also needed to integrate their influences. We will also include this in our benchmark as a future agenda.
> >
> >
> > **Q3. It would be much better if the authors could propose a simple and effective method after having the analysis on the existing methods. Based on the experiment results, what are the most important factor for addressing continual graph learning, especially for the class incremental setting? New insights for promoting the future research is somehow missing in the paper.**
> >
> > **A3.** Thanks for this suggestion. Based on the experimental results, the key factor for addressing continual graph learning concerns both maintaining the performance on previous tasks (stability) and keeping enough plasticity for learning new tasks (line 310-315, and line 318-327).
> >
> > Specifically, the regularization based methods can preserve the performance on previous tasks, but the performance on new tasks is negatively affected by the constraints. Methods that adapt well to new tasks (e.g. Bare models) would fail to preserve performance on previous tasks. Among all the baselines (except Joint), the memory replay based method (ER-GNN) strikes a better balance between the stability and the plasticity, since no constrain is added to the model parameters. Inspired by our observations, we are developing a new method based on the memory replay mechanism, which aims to incorporate the topological information of graphs (which has not been explicitly considered in ER-GNN, GEM, etc.). Our initial results have shown promising performance on Arxiv-CL and CoraFull-CL datasets, we will also include it in the benchmark later.
> >
> > Please let us know if there is any remaining concern, and we are more than happy to address them.

---

> ### Author Response · Authors · 2022-08-28
> **We are happy to address any remaining/further concern/question from Reviewer ynGE**
>
> We sincerely thank the reviewer for the recognition of our work and the valuable questions/suggestions in the reviews, which are very constructive to further improve our work, and we have made great efforts (including a number of additional experiments) to address them.
>
> Given the limited time for discussion, we would really appreciate if the reviewer could let us know whether we have made satisfying improvements with all these efforts, and if there is any further concern/question. We are more than happy to address any further concern/question.

---

### Official Review · Reviewer_d9V3 · 2022-07-26

**Rating:** 6
**Confidence:** 4
**Clarity:** This paper is well written.

**Strengths:**

- It’s an up-to-date benchmark that strives to provide fair comparisons for CGL methods, which is meaningful for the development of the related fields.
- The settings in this benchmark are elaborated by presenting the connections with the classical continual learning settings and demonstrating the graph-specific settings.
- Comprehensive evaluations are conducted based on 8 continual learning methods, and some insights across different datasets have been provided.
- The released toolkit is well-organized and easy to follow.


**Weaknesses:**

- A key concern is that the evaluation metric, i.e., performance matrix $M^p$ only considers sequences of two tasks, rather than a sequence of tasks stated in the section 3.1. Consequently, the experiments are limited in terms of continual learning settings.
- It’s would be better to discuss the difference between the current benchmark and the previous benchmark [1][2] in detail.
- The description “In contrast, CGL targets the forgetting problem, therefore the data from previous tasks are inaccessible. Few-shot graph learning aims at fast model adaptation to new tasks. In the training, few-shot learning models have access to all tasks simultaneously (not available in CGL).” in section 2.2 should be more careful, since N-CGL (with inter-task edges) proposed in this paper actually leverages the data from existing tasks.
- When dividing classes into tasks, K is supposed to be set as multiple choices to provide more insights.
- It’s weird that most continual learning methods fail to bring improvements in Arxiv-CL according to Table 5. The authors are expected to present additional analysis or demonstrate individual task performances.

[1] Antonio Carta, Andrea Cossu, Federico Errica, and Davide Bacciu. Catastrophic forgetting in deep graph networks: an introductory benchmark for graph classification. arXiv preprint arXiv:2103.11750, 2021.

[2] Stanley, Megan, et al. "Fs-mol: A few-shot learning dataset of molecules." Thirty-fifth Conference on Neural Information Processing Systems Datasets and Benchmarks Track (Round 2). 2021.


**Additional Feedback:**

As has been discussed:
- More evaluations and analysis are supposed to be presented.
- Some descriptions should be more careful.


**Correctness:**

The benchmark seems technologically sound and the related details are sufficient.

**Documentation:**

Generally, it provides sufficient details on data collection and organization, availability and maintenance, and ethical and responsible use.

**Ethics:**

I don't find any ethical concerns in this submission.

**Relation To Prior Work:**

Related works are properly discussed.

**Summary And Contributions:**

This paper studies continual learning scenarios on graph data (CGL), which have rarely been discussed under a unified framework previously. It generally categorizes the CGL into node-level and graph-level under task-incremental and class-incremental settings. Some graph-specific settings (e.g., inter-connection among tasks in N-CGL) are detailed to help understand the graph-specific properties in CGL. The authors have shown the attached analysis by implementing baselines and evaluating them on some public graph datasets. The corresponding datasets and training/evaluation toolkits are also provided.

---

> ### Author Response · Authors · 2022-08-11
> **Responses to Reviewer d9V3 (Part 2)**
>
> **Q3. The description “In contrast, CGL targets the forgetting problem, therefore the data from previous tasks are inaccessible. Few-shot graph learning aims at fast model adaptation to new tasks. In the training, few-shot learning models have access to all tasks simultaneously (not available in CGL).” in section 2.2 should be more careful, since N-CGL (with inter-task edges) proposed in this paper actually leverages the data from existing tasks.**
>
> **A3.** Thanks for this suggestion. We have carefully modified this statement to be more rigorous (line 116-119 and line 121-122). Specifically, we explained that CGL with inter-task edges allows access to previous information via the neighborhood aggregation of GNNs, but the access is restricted to the node features and the labels are inaccessible.
>
>
> **Q4. When dividing classes into tasks, K is supposed to be set as multiple choices to provide more insights.**
>
>
> **A4.** Thanks for this suggestion. We have now added a new section (Section 2.5 in Appendix) investigating the influence of K on the performance, and the experimental results with multiple different choices of K are included in Table 4 of Appendix (also shown below for convenience).
>
> Table: Performance comparisons under class-IL on Arxiv-CL with different task splittings ($\uparrow$ higher means better).
>
> | C.L.T.  | K=2  | | K=5 | | K=10 | | K=20| |
> | ---------| ------- |-|-------|-|---------|-|-------|-|
> | |AP/\% $\uparrow$ | AF/\% $\uparrow$ | AP/\% $\uparrow$ | AF/\% $\uparrow$ |AP/\% $\uparrow$ |AF/\% $\uparrow$|AP/\% $\uparrow$ |AF/\% $\uparrow$|
> | Bare model| 4.9$\pm$0.0|-88.4$\pm$0.3 | 10.5$\pm$0.1|-77.5$\pm$0.5 | 16.4$\pm$0.2|-63.9$\pm$0.6 | 26.4$\pm$0.3|-47.3$\pm$0.9|
> | EWC   |4.9$\pm$0.2|-89.2$\pm$0.4 | 9.4$\pm$0.1|-73.7$\pm$1.1 | 15.7$\pm$0.3|-62.8$\pm$0.7 | 24.8$\pm$0.3|-47.5$\pm$0.6|
> | MAS   |4.9$\pm$0.1|-87.0$\pm$0.2 | 10.3$\pm$0.2|-77.5$\pm$0.6 | 16.5$\pm$0.3|-64.0$\pm$0.5 | 26.3$\pm$0.6|-47.5$\pm$0.7|
> | GEM   |4.8$\pm$0.5|-87.8$\pm$0.2 | 10.7$\pm$0.1|-81.5$\pm$0.3 | 18.2$\pm$0.2|-70.6$\pm$0.5 | 31.3$\pm$0.1|-58.5$\pm$0.2|
> | TWP   |4.9$\pm$0.1|-89.4$\pm$0.3 | 8.3$\pm$0.4|-66.1$\pm$1.3 | 14.0$\pm$0.4|-57.6$\pm$1.5 | 22.0$\pm$0.4|-47.6$\pm$0.5|
> | LwF   |4.9$\pm$0.1|-87.5$\pm$0.2 | 24.2$\pm$0.4|-31.9$\pm$1.0 | 19.6$\pm$1.1|-41.8$\pm$1.7 | 19.6$\pm$0.7|-51.1$\pm$0.1|
> | ER-GNN|12.3$\pm$3.1|-79.9$\pm$3.3 | 10.9$\pm$0.2|-77.5$\pm$0.5 | 19.8$\pm$1.2|-59.9$\pm$1.3 | 31.6$\pm$0.6|-34.8$\pm$1.3|
> | Joint |56.8$\pm$0.3|- | 55.3$\pm$0.0|- | 53.9$\pm$0.0|- | 51.6$\pm$0.1|- |
>
> Initially, we set K=2 to maximize the number of tasks in the sequences, which increases the continual learning difficulty (line 165-167 in the paper). However, in our implemented pipeline, K can be easily modified by specifying the corresponding arguments in our implemented pipelines.
>
> As shown in the empirical results, the performance (AP and AF) decrease significantly when K becomes smaller. Therefore, we conclude that smaller K (longer sequences of tasks) magnifies the forgetting issue since the performance on each task is easier to be interfered with a larger number of tasks.
>
>
> **Q5. It’s weird that most continual learning methods fail to bring improvements in Arxiv-CL according to Table 5. The authors are expected to present additional analysis or demonstrate individual task performances.**
>
> **A5.** Thanks for this suggestion. This phenomenon was briefly analyzed in Section 4.2 (line 310-313). To further investigate this issue, in the revised version, we additionally visualized the performance matrices of all baselines on Arxiv-CL (Section 2.2 of Appendix, line 107-120), which show the methods' performance on every previous task after learning each new task.
>
> According to the results, we conclude that the failure of the baselines comes from the strong inter-task interference in Arxiv-CL. Specifically, according to the diagonal entries, all baselines obtain reasonable performance on individual tasks. However, the methods with severest forgetting on previous tasks (Bare model, EWC, GEM, and LwF) exhibit better performance when learning new tasks, and vice versa (TWP, ER-GNN, and MAS preserve the performance on previous tasks better, and their performance on the following tasks are lower). Moreover, the jointly trained models (Joint), which does not have forgetting issue, maintains a balanced performance on all tasks. But the diagonal entries of its performance matrix have lower values than the methods with severe forgetting. In other words, Joint cannot simultaneously perform well on all tasks, which further validates the strong task-wise interference in Arxiv-CL.
>
> Please let us know if there is any remaining concern, and we are more than happy to address them.

---

> ### Author Response · Authors · 2022-08-11
> **Responses to Reviewer d9V3 (Part 1)**
>
> We sincerely thank the reviewer for the recognition of our work and the constructive comments. Detailed responses are provided below.
>
>
> **Q1. A key concern is that the evaluation metric, i.e., performance matrix $M^p$ only considers sequences of two tasks, rather than a sequence of tasks stated in the section 3.1. Consequently, the experiments are limited in terms of continual learning settings.**
>
> **A1.** Thanks for raising this question. $M^p$ actually considers a sequence of $T$ tasks (the detailed numbers of tasks $T$ are provided in Table 1 and 2) instead of only sequences of two tasks. Specifically, $M^p\in \mathbb{R}^{T\times T}$, where $T$ is the length of the task sequence.
> **Each entry $M^p_{i,j}$ denotes the performance on task $j$ after the model has been trained over a sequence of tasks from 1 to $i$ (rather than just training the model only on task $i$ and getting the performance on task $j$).** We have added additional explanation on this to avoid confusion (line 208-209)
>
> Therefore, the $j$-th column of the matrix (i.e., {$M^p_{i,j}$, $i=j,...,T$}, $i$ doesn't start from 1 because $M^p$ is lower triangular) denotes how the model's performance on task $j$ varies after learning each new task $i$ ($i=j,...,T$).
> And the last ($T$-th) row of the matrix contains the model's performance on every learnt task after learning from task 1 to task $T$.
>
> In our experiments, we also visualized the performance matrix on task sequences of tens of tasks (Figure 4 in paper, Figure 1-5 in Appendix).
>
>
> **Q2. It’s would be better to discuss the difference between the current benchmark and the previous benchmark [1][2] in detail.**
>
> **[1] Antonio Carta, Andrea Cossu, Federico Errica, and Davide Bacciu. Catastrophic forgetting in deep graph networks: an introductory benchmark for graph classification. arXiv preprint arXiv:2103.11750, 2021.**
>
> **[2] Stanley, Megan, et al. "Fs-mol: A few-shot learning dataset of molecules." Thirty-fifth Conference on Neural Information Processing Systems Datasets and Benchmarks Track (Round 2). 2021.**
>
> **A2.** Thanks for this suggestion. The discussion on [1] can be found in Section 2.2, and we have enriched it with more details in the revised version (line 89-99). Specifically, [1]  has the following major differences compared to our CGLB.
>
> 1. [1] focuses on the graph-level tasks, while our CGLB contains both node-level and graph-level tasks.
>
> 2. [1] adopted three datasets. Only one of them is a graph dataset, while the other two are constructed from image datasets (MNIST and CIFAR10) by connecting the image pixels into grid graphs. CGLB is constructed based upon 7 different graph datasets which include citation networks, social networks, co-purchasing networks, and molecule graphs.
>
> 3. [1] adopted 4 baselines originally designed for traditional continual learning. In our work, we benchmarked 8 baselines including 6 popular continual learning approaches and 2 state-of-the-art continual graph learning approaches.
>
>
> We have also added detailed discussion on [2] in Section 2.2 of the paper (line 99-103). Specifically, [2] proposed a novel molecule dataset and a benchmarking procedure to facilitate the researches on few-shot graph learning. In contrast, the benchmark we proposed focuses on continual graph learning (CGL). As explained in Section 2.2 (line 119-124), few-shot graph learning and CGL are essentially different in terms of both the target and the experimental settings. Few-shot learning focuses on fast adaption to new tasks, while CGL focuses on reducing the forgetting issue. During training, few-shot learning allows access to all tasks simultaneously, while CGL only allow access to data of the current task (the scenario with the inter-task edges allows access to previous information through the neighborhood aggregation of GNNs, but the labels are inaccessible). Besides, [2] focuses on graph-level tasks on the molecule dataset, while we focus on both node-level and graph-level tasks on different datasets including citation network, social network, co-purchasing network, and molecules. In the future, we would like to incorporate this dataset into our benchmark.

---

> ### Author Response · Authors · 2022-08-28
> **We are happy to address any remaining/further concern/question from Reviewer d9V3**
>
> We sincerely thank the reviewer for the recognition of our work and the valuable questions/suggestions in the reviews, which are very constructive to further improve our work, and we have tried our best to address them.
>
> First, we provided detailed explanations to clarify the **key concern on the evaluation metric**. Second, we carefully adopted the suggestion to investigate different K with a number of additional experiments. Third, in the revised version, we added detailed discussion on the difference between our CGLB and the benchmark works mentioned by the reviewer. Fourth, regarding the concern on the failure of different baselines on Arxiv-CL under class-IL setting, we additionally provided more detailed experimental results and analyzed the underlying reason. Finally, we also carefully revised an expression on the experimental setting to be more rigours as advised by the reviewer.
>
> Given the limited time for discussion, we would really appreciate if the reviewer could let us know whether we have made satisfying improvements with all these efforts, and if there is any further concern/question. We are more than happy to address any further concern/question.

---

### Official Review · Reviewer_73qe · 2022-07-26
**Review for CGLB paper**

**Rating:** 7
**Confidence:** 3
**Clarity:** Yes, I think the paper is overall wel…

**Strengths:**

1. This work is overall well-motivated. Benchmarking the experimental settings and configuration is very essential in the ML area. Besides, a more difficult and practical task class-incremental setting can also benefit the continual graph learning area.
2. This work provides analyses and insights into the experimental results.
3. The submission is accompanied by a code repo containing the preprocessing and evaluating code.

**Weaknesses:**

1. The authors mention in line 255: "Due to the message passing mechanism of GNNs, the inter-task edges also break the restriction on the access to the information of the previous tasks". Since the message passing can leverage multi-hop neighbor information, a deeper GNN model can access more information from the previous tasks. I recommend the authors discuss the layer of the GNN model to make the comparison fairer.
2. The presened toolkit (GitHub repo) should contain a detailed document to make it more user-friendly.

**Additional Feedback:**

I don't have moe comments.

**Correctness:**

I think the authors' claim in this paper is correct, with a clear description of the dataset construction process and the experiment settings.

**Documentation:**

Yes, the description of dataset construction seems clear, and github repo with the toolkit is online.

**Ethics:**

No. The datasets are constructed based on the public datasets, I don't think there are any ethics issues in this work.

**Relation To Prior Work:**

This paper has a clear discussion of related work on continual learning and its application to graphs. I think the authors should discuss more about the relations between this work and existing datasets.

**Summary And Contributions:**

This work researches the experimental protocols in continual graph learning and develops the CGLB benchmark. The authors categorize continual graph learning tasks into node-level and graph-level, and both tasks contain task incremental and class incremental settings. This work also explores the state-of-the-art baselines on CGLB to benchmark the current progress in continual graph learning and release a toolkit for preprocessing the datasets.

---

> ### Author Response · Authors · 2022-08-11
> **Responses to Reviewer 73qe**
>
> We sincerely thank the reviewer for the recognition of our work and the constructive comments. Detailed responses are provided below.
>
> **Q1. The authors mention in line 255: "Due to the message passing mechanism of GNNs, the inter-task edges also break the restriction on the access to the information of the previous tasks". Since the message passing can leverage multi-hop neighbor information, a deeper GNN model can access more information from the previous tasks. I recommend the authors discuss the layer of the GNN model to make the comparison fairer.**
>
> **A1.** Thanks for this suggestion. In our experiments, we have adopted a 2-layer configuration for all different continual graph learning baselines to ensure a fair comparison.
>
> We have now strengthened this in both paper (line 249-250) and Appendix (line 59-60).
>
> **Q2. The presened toolkit (GitHub repo) should contain a detailed document to make it more user-friendly.**
>
> **A2.** Thanks for this suggestion. On our GitHub page, we have provided usage instructions with concrete examples for both the NCGL and GCGL tasks. After revision, we also added instructions on implementing new CGL methods using our toolkit, which is also added as a new Section in Appendix (Section 3).
>
> **Q3. I think the authors should discuss more about the relations between this work and existing datasets.**
>
> **A3.** Thanks for this suggestion. Discussions on related datasets and benchmarks can be found in Section 2.2 in the paper. In the revised version, we also enriched the original discussion with more detailed analysis and additional related datasets (Section 2.2, line 87-103).
>
> Please let us know if there is any remaining concern, and we are more than happy to address them.

---

> > ### Author Response · Authors · 2022-08-23
> > **Thank you for the further responses (Reviewer 73qe)**
> >
> > We sincerely thank the reviewer for recognizing our contribution, and we are more than happy to know that major concerns were successfully resolved. Moreover, we appreciate the reviewer's constructive suggestions on further improving our benchmarks which are our ongoing works and serve as a great complementary for our current work.
> >
> > **Q1. I encourage the authors to continue to improve the presented toolkit, construct a webpage, write detailed documentation.**
> >
> > 1. We have constructed a documentation webpage (https://cglb.readthedocs.io/en/latest/), which contains detailed documentation for all implemented baselines and the utility functions for evaluation and visualization. Concrete examples are also provided. The webpage is complementary to our GitHub repository and makes it easier for other researchers to follow our work.
> >
> > **Q2. Make it support heterogeneous graphs (if possible) to make this work easy to follow.**
> >
> > 2. Supporting heterogeneous graphs is very beneficial for the development of the continual graph learning community, and we are working on the implementations. First, we are now constructing benchmark tasks from several widely adopted data sources for heterogeneous graphs, including DBLP (academic network), IMDB (film rating network), Yelp (social media network), Amazon (E-commercial network). IMDB and Amazon are obtained via *https://grouplens.org/datasets/movielens/100k/* and *http://jmcauley.ucsd.edu/data/amazon/*. While the curation of DBLP and Yelp follows the process proposed in a heterogeneous graph benchmark work [1]. The task construction will follow the node-level continual graph learning (NCGL) task construction proposed in our paper, with both task-IL and class-IL scenarios. Second, to implement the continual learning baselines on heterogeneous graphs, heterogeneous GNNs will have to be implemented as the backbones. Several representative heterogeneous GNNs including Heterogeneous Graph Neural Network (HetGNN) [2], Heterogeneous Graph Attention Network
> > (HAN) [3], Heterogeneous Graph
> > Transformer (HGT) [4], Relational Graph Convolutional Networks (R-GCNs) [5], Metapath2vec [6], and Predictive Text Embedding (PTE) [7] will be implemented into our continual learning pipelines, which will be benchmarked with the constructed tasks. Both the constructed benchmark tasks and the implemented baselines will be included in our CGLB upon completion. The corresponding discussion is also added to *Appendix (Section 5, line 182-198).*
> >
> > Again, we sincerely thank the reviewer for the recognition of our contribution and the constructive suggestions. Please let us know in case there is any further concern, and we are more than happy to address them.
> >
> > [1] Yang, Carl, et al. "Heterogeneous network representation learning: A unified framework with survey and benchmark." IEEE Transactions on Knowledge and Data Engineering (2020).
> >
> > [2] Zhang, Chuxu, et al. "Heterogeneous graph neural network." Proceedings of the 25th ACM SIGKDD international conference on knowledge discovery & data mining. 2019.
> >
> > [3] Wang, Xiao, et al. "Heterogeneous graph attention network." The world wide web conference. 2019.
> >
> > [4] Hu, Ziniu, et al. "Heterogeneous graph transformer." Proceedings of The Web Conference 2020. 2020.
> >
> > [5] Schlichtkrull, Michael, et al. "Modeling relational data with graph convolutional networks." European semantic web conference. Springer, Cham, 2018.
> >
> > [6] Dong, Yuxiao, Nitesh V. Chawla, and Ananthram Swami. "metapath2vec: Scalable representation learning for heterogeneous networks." Proceedings of the 23rd ACM SIGKDD international conference on knowledge discovery and data mining. 2017.
> >
> > [7] Tang, Jian, Meng Qu, and Qiaozhu Mei. "Pte: Predictive text embedding through large-scale heterogeneous text networks." Proceedings of the 21th ACM SIGKDD international conference on knowledge discovery and data mining. 2015.

---

### Official Review · Reviewer_H68Y · 2022-07-28
**Continual Graph Learning**

**Rating:** 5
**Confidence:** 3
**Correctness:** Seems correct
**Clarity:** Yes

**Strengths:**

- Graph Learning is an interesting topic
- Results seem reasonable

**Weaknesses:**

- It seems like it doesnt fit the dataset track as its more like a technique from my understanding.
- From the benchmarking side of view its unclear why this method should be the benchmark (explanation or clear differentation is missing)

**Additional Feedback:**

None

**Documentation:**

Good

**Ethics:**

No issues

**Relation To Prior Work:**

Yes.

**Summary And Contributions:**

CGLB, provide a comprehensive toolkit for training, evaluation, and visualization of different CGL
models, which could greatly alleviate the burden/risks to investigate CGL problems. Finally, with
CGLB and the toolkit, we conduct comprehensive studies on the existing state-of-the-art methods,
which reveal the effectiveness of these methods in different scenarios and point out challenges of the
existing models under certain settings.

---

> ### Author Response · Authors · 2022-08-11
> **Responses to Reviewer H68Y**
>
> We sincerely thank the reviewer for raising these concerns. Detailed responses are provided below.
>
> **Q1. It seems like it doesnt fit the dataset track as its more like a technique from my understanding. From the benchmarking side of view its unclear why this method should be the benchmark (explanation or clear differentation is missing)**
>
> **A1.** Thanks for this concern. Besides the comprehensive toolkit recognized by the reviewer, our key contribution is the novel and comprehensive benchmark for continual graph learning (CGLB). This is also the first continual graph learning benchmark which covers both node-level and graph-level tasks in both task-IL and class-IL scenarios.
>
> Our contributions from the benchmark perspective can be found in both Abstract and Introduction. Specifically, as explained in Abstract (line 9-14), ''we systematically study the task configurations in different application scenarios and develop a comprehensive Continual Graph Learning Benchmark (CGLB) curated from different public datasets. Specifically, CGLB contains both node-level and graph-level continual graph learning tasks under task-incremental (currently widely adopted) and class-incremental (more practical, challenging, yet underexplored) settings''.
> As explained in Introduction (line 42-49), ''In this paper, we first provide a systematic taxonomy of different CGL scenarios that include (1) node-level CGL (N-CGL), which deals with node-level prediction on a single growing graph with new types of nodes continuously emerging; and (2) graph-level CGL (G-CGL), which deals with graph-level prediction with new categories of graphs continuously appear. Moreover, we clarify the difference between the currently widely adopted task-incremental learning (task-IL) setting and the underexplored yet more challenging class-incremental learning (class-IL) setting, and point out the proper application scenarios of these two settings. Based on the established taxonomy, we construct the Continual Graph Learning Benchmark (CGLB) with benchmark tasks for each setting.''
>
> Our contribution from the benchmark perspective is also recognized by the other reviewers, e.g. ''This work proposes a novel benchmark for continual graph learning.'' (Reviewer 5sBj), ''This work is overall well-motivated. Benchmarking the experimental settings and configuration is very essential in the ML area.'' (Reviewer 73qe), ''It’s an up-to-date benchmark that strives to provide fair comparisons for CGL methods, which is meaningful for the development of the related fields.'' (Reviewer d9V3), etc..
>
> Please let us know if there is any remaining concern, and we are more than happy to address them.

---

> > ### Comment · Reviewer_H68Y · 2022-08-15
> > **Unclear**
> >
> > Thank you for your response.  The next time consider that the reviewer read your paper and did not get it. So just citing things does not help my understanding, as well citing other reviewers is useless for my understanding.
> >
> > So let me make two things very clear:
> > 1. ) What is the difference against standard Frameworks like :
> > https://deepai.org/publication/continual-graph-learning
> > GPN: A Joint Structural Learning Framework for Graph Neural Networks
> > https://deepai.org/publication/gpn-a-joint-structural-learning-framework-for-graph-neural-networks
> > A Unified Continuous Learning Framework for Multi-modal Knowledge Discovery and Pre-training
> > https://arxiv.org/abs/2206.05555
> >
> > 2.) I tried to run your code but it gives multiple errors. Please enlighten me how to install it correctly.
> > "Get Started i did"
> >
> > But yet the command does not run on my end, as I guess it is Linux only dependent?
> > " OSError: [Errno 22] Invalid argument: "./results/errors/no_inter_task_edges/tsk_IL/train_ratio_0.2/Products-CL_2_bare_[None]_GCN_{'h_dims': [256], 'dropout': 0.0, 'batch_norm': False}_True_True_200_1.pkl" "

---

> > > ### Author Response · Authors · 2022-08-16
> > > **Further responses to Reviewer H68Y (Part 2)**
> > >
> > > **Q2. About running the code (in Windows system)**
> > >
> > > **A2.** Yes, the code originally only supported Linux system (our system is Ubuntu 18). However, to facilitate Windows users, we upgraded our code to resolve the error encountered by the reviewer.
> > > Specifically, the error message (invalid argument) provided by the reviewer is caused by illegal filename characters like '\{', '[', ':', ',', etc.. And we have reproduced this error in Windows system. To circumvent this problem, we implemented a new function to replace potential illegal filename characters with the underscore symbol "_", which has been updated into our GitHub repository. With the latest code, users can simply specify the argument ```--replace_illegal_char``` as ```True``` to enable this function. For example,
> > >
> > > python train.py --dataset Arxiv-CL \ \
> > >        --method bare \ \
> > >        --backbone GCN \ \
> > >        --gpu 0 \ \
> > >        --ILmode taskIL \ \
> > >        --inter-task-edges False \ \
> > >        --minibatch False \ \
> > >        --**replace\_illegal\_char True** \
> > >
> > > The corresponding instructions are also added on our GitHub page at the beginning.
> > >
> > > Besides, to facilitate users with only CPUs, we also implemented a CPU version of our code, which is available via [https://github.com/QueuQ/CGLBcpu](https://github.com/QueuQ/CGLBcpu).
> > >
> > > We have tested the code in Windows system on both GPUs and CPUs, and the code can successfully run.
> > >
> > > Again, we sincerely thank the reviewer for further clarifying the concerns so that we have a chance to further clarify and improve our work. Please let us know in case there is any further concern, and we are more than happy to address them.

---

> > > ### Author Response · Authors · 2022-08-16
> > > **Further responses to Reviewer H68Y (Part 1)**
> > >
> > > We really appreciate the reviewer took time to further clarify the concerns, which are very constructive and helpful for us to further improve the manuscript. Also apologize for misunderstanding the questions earlier.
> > >
> > > Our detailed responses to the concerns are listed below:
> > >
> > > **Q1. What is the difference against standard Frameworks like : https://deepai.org/publication/continual-graph-learning GPN: A Joint Structural Learning Framework for Graph Neural Networks https://deepai.org/publication/gpn-a-joint-structural-learning-framework-for-graph-neural-networks A Unified Continuous Learning Framework for Multi-modal Knowledge Discovery and Pre-training https://arxiv.org/abs/2206.05555**
> > >
> > > **A1.**
> > > Briefly speaking, there are two major differences between our work and these three works. First, these three works focus on developing specific novel methods for certain problems. Instead of directly developing novel methods, we aim to facilitate the development of novel methods for the continual graph learning (CGL) problem via our proposed CGLB, which contains standarized benchmark tasks and evaluation protocols to fairly compare different CGL methods. Besides, we also provide a systematic study on the performance of existing CGL methods with analysis on their characteristics. Therefore, our contribution is from the benchmark perspective, while their contributions are from the method perspective. Second, "GPN: A Joint Structural Learning Framework for Graph Neural Networks" and "A Unified Continuous Learning Framework for Multi-modal Knowledge Discovery and Pre-training" do not focus on the CGL problem studied in our work, and their experiments are not under the continual learning settings. Detailed comparisons with each of these works are listed below. We have also cited all these papers in the revised version.
> > >
> > > **1. Comparison between our CGLB and the work 'Continual Graph Learning' (https://deepai.org/publication/continual-graph-learning).** First, the work 'Continual Graph Learning' proposed a novel method for CGL in the task-incremental (task-IL) scenario with node-level tasks (contributing from the model structure perspective). In contrast, we proposed a set of standard benchmark tasks (CGLB) to fairly compare different methods in both task-IL and class-IL scenarios with both node-level and graph-level tasks (contributing from the benchmark perspective). Second, the methods in the experiments of 'Continual Graph Learning' included only one continual learning method (the ER-GNN they proposed), and the baselines are bare GNNs without any continual learning technique. In our work, we included 8 continual learning baselines including 2 baselines specially designed for CGL. Third, 'Continual Graph Learning' adopted 3 datasets for node-level tasks. In our work, we adopted 7 datasets including 4 datasets for node-level tasks and 3 for graph-level tasks. Our adopted node-level datasets also included large ones with millions of nodes. Finally, we provide a comprehensive toolkit including modularized pipelines to facilitate the development of new CGL methods, which is not the target of 'Continual Graph Learning'.
> > >
> > >
> > > **2. Comparison between our CGLB and the work 'GPN: A Joint Structural Learning Framework for Graph Neural Networks' (https://deepai.org/publication/gpn-a-joint-structural-learning-framework-for-graph-neural-networks).** GPN is a novel method to handle graph data with potential missing edges, which operates under the standard learning setting, instead of the continual learning setting with the catastrophic forgetting issue. In contrast, our novel benchmark (CGLB) aims to facilitate the evaluation and development of CGL methods operating under continual learning settings, and is highly focused on benchmarking the forgetting issue of different methods. In other words, we target the continual graph learning (CGL) problem, and our contribution is a novel benchmark. While GPN targets a specific but different learning problem, and its contribution is a specific novel method.
> > >
> > > **3. Comparison between our CGLB and the work "A Unified Continuous Learning Framework for Multi-modal Knowledge Discovery and Pre-training" (https://arxiv.org/abs/2206.05555).** The work "A Unified Continuous Learning Framework for Multi-modal Knowledge Discovery and Pre-training" innovatively integrated knowledge discovery into the knowledge guided multi-modal pre-training so that the explicitly labeled knowledge graph is no longer needed. This work targets the multi-modal pre-training problem instead of CGL, and the pre-trained model is eventually applied to the image-text retrieval task instead of graph data related tasks. In contrast, our proposed benchmark (CGLB) focuses on CGL, and our implemented baseline models are applied to graph data related tasks (node-level and graph-level classification) under continual learning settings.

---

> ### Author Response · Authors · 2022-08-26
> **We are happy to address any remaining/further concern/question.**
>
> We sincerely thank the reviewer for the questions in the initial reviews and the further clarification. These questions (1. the difference between CGLB and several other works. 2. encountering error when running the code in Windows) are very constructive and valuable to further improve our work.
>
> For the first question, we have provided detailed explanations on the difference between our CGLB and each of the works mentioned by the reviewer.
>
> For the second question, we have updated our code to support Windows system with detailed instructions (after reproducing the error and confirming that the error was caused by the different filename rules in Windows). The updated code also supports both CPUs and GPUs.
>
> Given the limited time for discussion, we would really appreciate if the reviewer could let us know if the concerns are resolved, and if there is any remaining/additional concern/question. We are more than happy to address any remaining/further concern/question.

---

> ### Author Response · Authors · 2022-08-28
> **A friendly reminder to Reviewer H68Y that the discussion ends in one day**
>
> We sincerely thank the reviewer for the constructive reviews, and we have made great efforts to address the concerns.
>
> Given that the discussion is ending in only one day, we would really appreciate if the reviewer could let us know if the concerns are resolved, and if there is any remaining/additional concern/question. We are more than happy to address any remaining/further concern/question.

---

### Official Review · Reviewer_5sBj · 2022-07-31
**A good benchmark for continual graph learning**

**Rating:** 6
**Confidence:** 3

**Strengths:**

- *Presentation.* The paper is well-written and its main idea is easy to flow.

- *Benchmark.* The benchmark is well-designed and it addresses an important topic in graph learning with significant real-world application.

- *Analysis of existing methods.* Using the proposed benchmark and toolkit, the authors perform a systematic comparison of existing CGL methods.


**Weaknesses:**

- The benchmark contains implementation of a variety of existing CGL methods. However, it does not properly document how to implement and adapt new CGL methods to use this benchmark. This would limit the usability of the benchmark and the companying toolkit.


**Additional Feedback:**

---

**Clarity:**

The paper is well-written. The main idea is easy-to-follow.


**Correctness:**

The paper does not have apparent technical errors. The claims are supported by experiments.


**Documentation:**

- The design of the benchmark tasks and the datasets are well-explained.

- The steps to run the pipeline in the toolkit is well-documented. However, how to extend the pipeline with more methods is not mentioned.

- The instructions to reproduce the results in the paper are well-documented.


**Ethics:**

---

**Relation To Prior Work:**

The related work is comprehensively discussed. The authors not only relate their work to continual learning, but also compares it with existing benchmarks for CGL.


**Summary And Contributions:**

This work proposes a novel benchmark for continual graph learning. The benchmark factorizes continual graph learning tasks into node- and graph-level tasks, and task-incremental and class-incremental settings. They also releases a toolkit for training, evaluating and visualizing CGL methods. Finally, the authors performs a systematic comparison of state-of-the-art CGL methods.

Contributions:

- A novel and comprehensive benchmark for continual graph learning methods.

- An easy-to-use toolkit for training, evaluating and visualizing CGL methods.

- A systematic comparison of existing CGL methods.

---

> ### Author Response · Authors · 2022-08-11
> **Responses to Reviewer 5sBj**
>
> We sincerely thank the reviewer for the recognition of our work and the constructive suggestions. Detailed responses are provided below.
>
> **Q1. The benchmark contains implementation of a variety of existing CGL methods. However, it does not properly document how to implement and adapt new CGL methods to use this benchmark. This would limit the usability of the benchmark and the accompanying toolkit.**
>
> **A1.** Thanks for the constructive suggestion.
> Our implemented pipelines for both node-level and graph-level methods are highly modularized, and new methods can be easily integrated.
>
> Specifically, implementation of a new method ''New\_NCGL'' for node-level tasks (contained in a New\_NCGL\_model.py file) can be placed under the *CGLB/NCGL/Baselines* directory, which also contains the implementations of all the baselines reported in our paper. Then the new method ''New\_NCGL'' is integrated in the pipeline for training and testing using our benchmark tasks. Similarly, implementations of new methods for graph-level tasks can be added to the directory *CGLB/GCGL/Baselines*.
>
> We have now added detailed explanations with concrete examples on implementing new methods in a new section of Appendix (Section 3), as well as on our GitHub page.
>
> Please let us know if there is any remaining concern, and we are more than happy to address them.

---

> > ### Author Response · Authors · 2022-08-23
> > **Further updates to support implementations of new methods**
> >
> > As a complement to the instructions on new method implementation provided in our initial responses, our newly constructed documentation webpage (https://cglb.readthedocs.io/en/latest/) can further facilitate the implementations. Specifically, the improvements are from two aspects.
> >
> > 1. We added a code template (in the form of a python class) for new method implementations (https://cglb.readthedocs.io/en/latest/api.html#module-NCGL.Baselines.New_NCGL_model). Detailed explanations are also provided from different perspectives like the input and output formats, etc..
> >
> > 2. We also added documentation for each implemented baseline method reported in our paper (https://cglb.readthedocs.io/en/latest/api.html). Since all these baselines follow the template mentioned above, users can also refer to these concrete examples when implementing new methods.

---

> ### Author Response · Authors · 2022-08-28
> **We are happy to address any remaining/further concern/question from Reviewer 5sBj**
>
> We sincerely thank the reviewer for the recognition of our contribution and the constructive suggestion on documenting how to implement and adapt new CGL methods to use our benchmark, which is very helpful to further improve our work.
>
> Regarding this valuable suggestion, we have made great effort to facilitate the implementation of new CGL methods. We not only added detailed instructions on our GitHub page, but also developed a new documentation webpage. The documentation webpage includes a template for new method implementations and detailed documentations on the implementations of all CGL methods reported in our paper, which could be very helpful references when implementing new methods.
>
> Given the limited time for discussion, we would really appreciate if the reviewer could let us know if the concern on implementing new methods with our CGLB is resolved, and if there is any remaining/additional concern/question. We are more than happy to address any remaining/further concern/question.

---

### Meta-Review · Area_Chair_biQP · 2022-09-04

**Recommendation:** Accept
**Confidence:** 4

**Metareview:**

This paper had 7 reviews. The reviewers’ comments were overall favorable from the point of view of quality, clarity (i.e., the paper is well-written and its main idea is easy to flow’), originality (this work proposes a novel benchmark for continual graph learning), and significance (a benchmark for continual graph learning (CGLB) is an important contribution). Overall, I believe the authors have addressed the reviewers’ comments satisfactorily, including comments from H68Y. There are no issues with Correctness, Documentation or Ethics. The authors have included a GitHub repository to support reproducibility of their work with detailed instructions

---

### Decision · Program_Chairs · 2022-09-16

Accept